# Structure of alpha-synuclein fibrils derived from human Lewy body dementia tissue

Dhruva D. Dhavale [1,13], Alexander M. Barclay[2,13], Collin G. Borcik[3], Katherine Basore[4], Deborah A. Berthold[5], Isabelle R. Gordon[1], Jialu Liu[1], Moses H. Milchberg[3], Jennifer Y. O'Shea [1], Michael J. Rau [4], Zachary Smith[1], Soumyo Sen[6], Brock Summers[4], John Smith[7], Owen A. Warmuth[3], Richard J. Perrin [1,8], Joel S. Perlmutter [1,9], Qian Chen [7], James A. J. Fitzpatrick [4], Charles D. Schwieters[10], Emad Tajkhorshid [6], Chad M. Rienstra [2,3,11,12,14] ✉ & Paul T. Kotzbauer[1,14] ✉

The defining feature of Parkinson disease (PD) and Lewy body dementia (LBD) is the accumulation of alpha-synuclein (Asyn) fibrils in Lewy bodies and Lewy neurites. Here we develop and validate a method to amplify Asyn fibrils extracted from LBD postmortem tissue samples and use solid state nuclear magnetic resonance (SSNMR) studies to determine atomic resolution structure. Amplified LBD Asyn fibrils comprise a mixture of single protofilament and two protofilament fibrils with very low twist. The protofilament fold is highly similar to the fold determined by a recent cryo-electron microscopy study for a minority population of twisted single protofilament fibrils extracted from LBD tissue. These results expand the structural characterization of LBD Asyn fibrils and approaches for studying disease mechanisms, imaging agents and therapeutics targeting Asyn.

Parkinson disease (PD) is diagnosed clinically based on motor signs that include tremor, bradykinesia, rigidity and impaired postural reflexes. PD is defined pathologically by the accumulation of alpha-synuclein (Asyn) fibrils in neuronal cytoplasmic and neuritic inclusions known as Lewy bodies (LBs) and Lewy neurites (LNs)[1]. The role of Asyn in the pathogenesis of PD is supported by the identification of dominant mutations in the gene encoding Asyn (*SNCA*) in rare familial versions of PD[2–9]. Dementia occurs frequently in PD and is associated with pathologic Asyn accumulation in neocortex[10–12]. Dementia sometimes begins at approximately the same time as motor symptoms (Dementia with Lewy bodies or DLB), or up to 20 years after motor symptoms begin (PD with dementia or PDD). Lewy body dementia (LBD) encompasses this spectrum of clinical presentations.

[1]Department of Neurology and Hope Center for Neurological Disorders, Washington University School of Medicine, St. Louis, MO 63110, USA. [2]Center for Biophysics and Quantitative Biology, University of Illinois at Urbana-Champaign, Urbana, IL 61801, USA. [3]Department of Biochemistry, University of Wisconsin-Madison, Madison, WI 53706, USA. [4]Center for Cellular Imaging, Washington University School of Medicine, St. Louis, MO 63110, USA. [5]Department of Chemistry, University of Illinois at Urbana-Champaign, Urbana, IL 61801, USA. [6]Theoretical and Computational Biophysics Group, NIH Resource for Macromolecular Modeling and Visualization, Beckman Institute for Advanced Science and Technology, Department of Biochemistry, and Center for Biophysics and Quantitative Biology, University of Illinois at Urbana-Champaign, Urbana, IL 61801, USA. [7]Department of Materials Science and Engineering, University of Illinois at Urbana-Champaign, Urbana, IL 61801, USA. [8]Department of Pathology & Immunology, Washington University School of Medicine, St. Louis, MO 63110, USA. [9]Department of Radiology, Neuroscience, Physical Therapy and Occupational Therapy, Washington University School of Medicine, St. Louis, MO 63110, USA. [10]Computational Biomolecular Magnetic Resonance Core, National Institute of Diabetes and Digestive and Kidney Diseases, National Institutes of Health, Bethesda, MD 20892, USA. [11]Morgridge Institute for Research, University of Wisconsin-Madison, Madison, WI 53706, USA. [12]National Magnetic Resonance Facility at Madison, University of Wisconsin-Madison, Madison, WI 53706, USA. [13]These authors contributed equally: Dhruva D. Dhavale, Alexander M. Barclay. [14]These authors jointly supervised this work: Chad M. Rienstra, Paul T. Kotzbauer ✉e-mail: crienstra@wisc.edu; kotzbauerp@wustl.edu

Analysis of Asyn fibril structure in LBD guides the identification of high affinity ligands needed to develop a Positron Emission Tomography (PET) imaging agent that can quantify the accumulation of Asyn fibrils in living individuals. An Asyn imaging agent would improve diagnosis and provide a biomarker for tracking disease progression. It would also enable assessment of target engagement for therapeutic strategies targeting Asyn[13]. Structural analysis of Asyn fibrils may also provide insight into disease mechanisms, including (1) promotion of nucleation, growth and clearance of fibrils; and (2) regulation of cell-to-cell transfer that underlies disease progression.

Different Asyn fibril polymorphs are produced by incubating human recombinant monomeric Asyn protein under different conditions to induce fibril formation, including differences in pH, buffers and salts, as demonstrated by SSNMR[14–21] and cryo-EM studies[22–36]. The structure of Asyn fibrils isolated from postmortem multiple system atrophy (MSA) brain tissue, where fibrils accumulate in oligodendroglial cells, has been determined by cryo-EM[37]. The fold of MSA Asyn fibrils is distinct from in vitro forms, although it does share one structural element, a Greek key fold, with a subset of the reported in vitro forms[16,37]. LBD Asyn fibril structure has been difficult to analyze with cryo-EM because of the very low twist present in fibrils isolated from tissue. Yang et al. recently reported a 2.2 Å structure (PDB 8A9L) of Asyn fibrils from a minority population (25%) of LBD fibrils found to have high twist and to be amenable to cryo-EM analysis after extraction from LBD tissue[38]. The fold of these single protofilament LBD fibrils, termed the 'Lewy fold', is distinct from structures previously reported for MSA and for Asyn fibrils assembled in vitro.

To analyze LBD fibril structure by SSNMR, we developed a method to amplify fibrils extracted from postmortem LBD brain tissue using isotopically labeled human recombinant Asyn. We utilized seeding properties in a cell culture system to guide method development by comparing the properties of amplified fibrils with tissue-derived fibrils.

The amplification method enabled us to prepare multiple milligram quantities of amplified fibrils, which were used for structure determination by SSNMR. LBD amplified fibrils comprise a mixture of single protofilament and two protofilament fibrils with very low twist. The protofilament fold of the LBD amplified fibrils is highly similar to the fold of the single protofilament fibrils with high twist analyzed by single-particle cryo-EM.

## Results

### Production of amplified fibrils from LBD and MSA postmortem tissue

Previous studies of fibril growth for WT and mutant Asyn fibrils indicate that fibril structure is templated during growth of fibril seeds in the presence of recombinant Asyn monomer[39]. To produce fibrils that replicate the structure present in LBD tissue, we developed a protocol to isolate fibrils from brain tissue samples in an insoluble fraction and use them as seeds to grow fibrils with recombinant human Asyn protein produced in *E. Coli*. This amplification protocol utilizes multiple cycles of sonication-mediated fragmentation followed by quiescent incubation for 2–3 days (Fig. 1a).

We used a highly sensitive radioligand binding assay to quantify LBD fibril amplification and guide the optimization of the protocol. We measured the binding site density ($B_{max}$ = 42 pmol/nmol) and affinity ($K_d$ = 5 nM) of the radioligand for LBD amplified fibrils and used this information to convert measurements of bound radioligand in binding assays to fibril concentration (µM)[40,41]. In addition to high sensitivity, the radioligand assay provides high specificity for fibrils over monomer, which improved the precision of measurements for samples with a low concentration of fibrils and high concentration of monomer. An important goal was to optimize amplification from fibril seeds isolated from LBD tissue samples, while minimizing the accumulation of fibrils with insoluble fractions prepared from control tissue samples, which

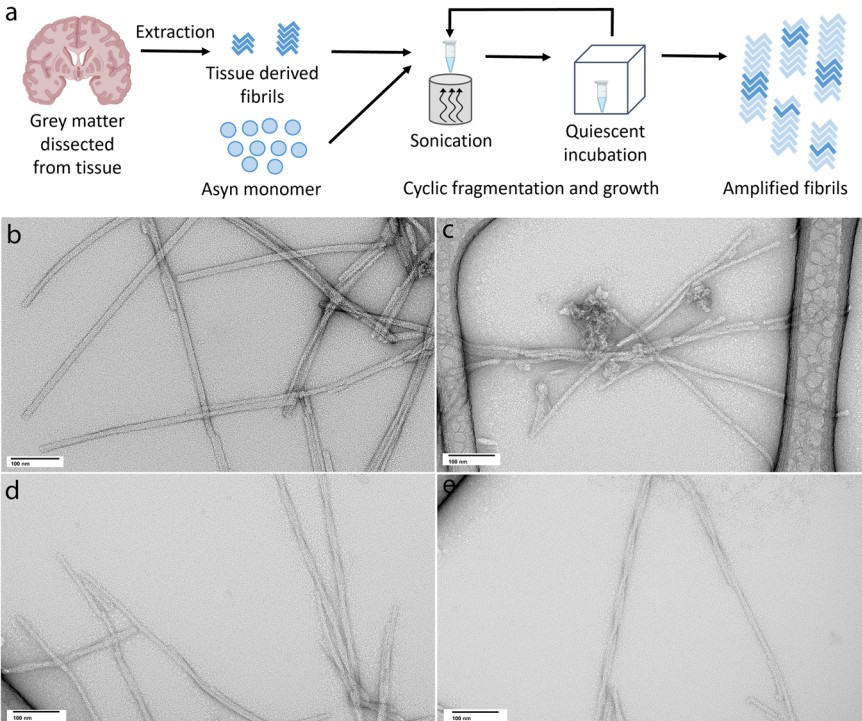

**Fig. 1 | Schematic illustrating extraction and amplification of LBD Asyn fibrils derived from postmortem tissue samples. a** Fibrils were isolated in an insoluble fraction by sequential extraction and centrifugation of tissue samples. The insoluble fraction seeds were combined with recombinant Asyn monomer and subjected to multiple rounds of sonication followed by quiescent incubation at 37 °C to generate amplified fibrils. **b–e** Representative negative stain TEM images of LBD amplified fibrils (**b**), control amplified fibrils (**c**), MSA amplified fibrils (**d**) and IV$_{Tris}$ fibrils (**e**). Similar results were obtained from negative stain TEM images collected for at least three independently prepared fibril samples.

were selected based on the absence of pathologic Asyn as determined by immunohistochemistry.

LBD amplification was initially optimized with natural abundance Asyn monomer. After 6 cycles, we observed 16-fold to 35-fold greater mass of fibrils amplified from LBD samples compared to control samples (Supplementary Table 1). Based on yield, we estimated that a minimum of 6 cycles of amplification were needed for structural studies. We further optimized amplification in the presence of isotopically labeled Asyn monomer to facilitate SSNMR studies and selected case LBD1 for large-scale amplification. We observed an 11-fold and 19-fold greater mass of LBD1 fibrils compared to two control cases (Supplementary Tables 2, 3). We used a micro-BCA assay and $A_{280}$ absorbance measurements to further evaluate fibril concentration and confirm the specificity of amplification for LBD compared to control samples after 6 cycles (Supplementary Tables 4, 5, 6). We obtained a yield of 5 mg and 4.2 mg of LBD1 fibrils amplified with uniform [$^{13}$C, $^{15}$N] labeled Asyn monomer (uCN) and with uniform [$^{13}$C, $^{2}$H, $^{15}$N] labeled Asyn monomer (uCDN) respectively, which were utilized for SSNMR studies (Supplementary Table 4).

Negative-stain transmission EM (TEM) analysis of amplified LBD fibrils showed straight fibrils with diameter range of 6–15 nm and no visible twist (Fig. 1b, Supplementary Fig. 1–3). Fibrils amplified from control tissue samples had smaller diameter and more curvature (Fig. 1c, (Supplementary Fig. 4)). As an approach to further evaluate structural fidelity of amplification, we also amplified fibrils from MSA tissue samples, which appear twisted (Fig. 1d, (Supplementary Fig. 5)). We also compared the morphology of amplified fibrils to those assembled in vitro by incubation of recombinant Asyn monomer at 37 °C with continuous shaking[39,41–47](Supplementary Fig. 6). For these fibrils, designated $IV_{Tris}$, the appearance of two twisting protofilaments was more clearly delineated by negative stain TEM than it was for the various forms of amplified fibrils. (Fig. 1e).

## Characterization of amplified fibrils by seeding Asyn aggregation in HEK293 cells – support for the structural fidelity of fibril amplification

We previously used a biosensor cell line to evaluate seeding activity of fibrils extracted from postmortem brain tissue samples and observed distinct seeding properties for LBD and MSA tissue[47]. We seeded these biosensor cells with fibrils amplified from either LBD or MSA tissue samples and compared their seeding properties to fibrils extracted directly from LBD and MSA tissue. We observed a consistent morphologic difference for intracellular inclusions produced by seeding with LBD and MSA amplified fibrils (after 4 cycles) in HEK293 cells (Fig. 2a–d). Inverted fluorescence microscopy images show that inclusions seeded with LBD amplified fibrils consistently have a more compact, clumped appearance. In contrast, inclusions seeded with MSA amplified fibrils contain long filamentous strands that expand throughout the cytoplasm. This distinct morphological appearance recapitulates the morphology produced by seeding with insoluble fractions obtained from LBD and MSA tissue samples (Fig. 2)[47].

Amplified LBD fibrils were substantially less efficient in seeding the biosensor cells compared to amplified MSA fibrils, consistent with our previous observations for insoluble fractions from LBD and MSA tissue (Supplementary Table 7). Our previous studies also demonstrated seeding activity in both soluble and insoluble fractions from MSA tissue, whereas seeding activity is detected only in the insoluble fractions from LBD tissue (Fig. 2e, f)[47]. We observed similar properties for amplified fibrils. Seeding activity was observed in the 100,000 x g supernatant for MSA amplified fibrils but not for LBD amplified fibrils. (Fig. 2g, h).

## Further characterization of fibril amplification by negative stain TEM, SDS-PAGE and protease digestion

Diameter measurements for LBD amplified fibrils indicated a bimodal distribution after 6 cycles of amplification, with diameters ranging

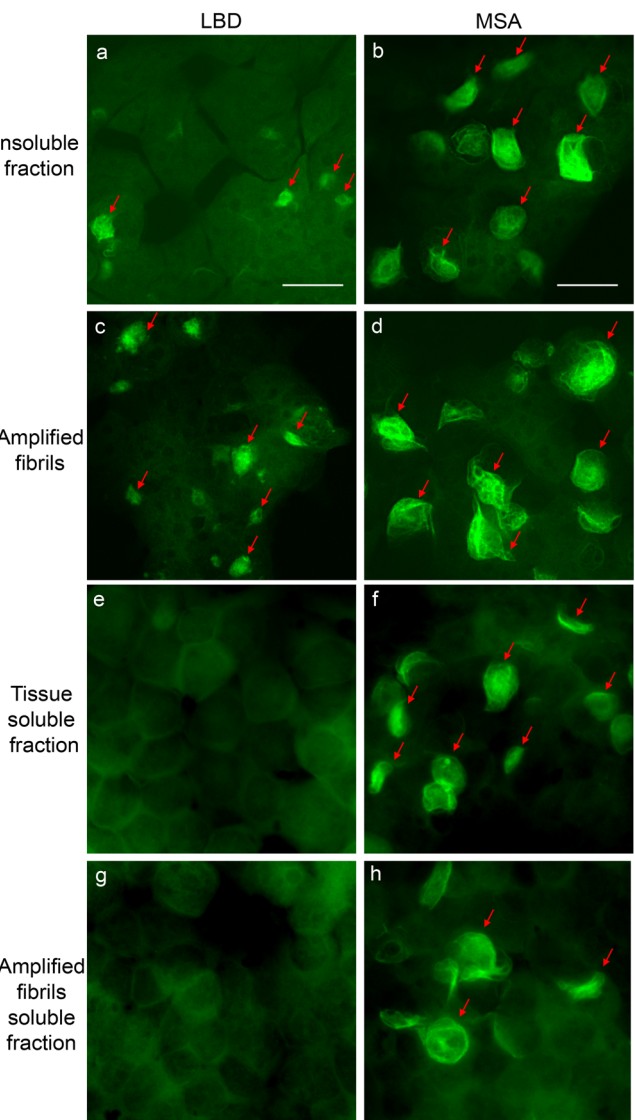

**Fig. 2 | Seeding properties of tissue-derived and amplified fibrils in HEK 293 biosensor cells expressing A53T Asyn-CFP/YFP.** Fluorescence images show cells seeded with: **a** LBD Insoluble fraction. **b** MSA insoluble fraction. **c** LBD amplified fibrils. **d** MSA amplified fibrils. **e** LBD tissue soluble fraction. **f** MSA tissue soluble fraction. **g** LBD amplified fibrils soluble fraction. **h** MSA amplified fibril soluble fraction. Red arrows indicate intracellular inclusions. Similar results were observed in more than three independent experiments examining amplified fibrils for three cases of LBD and three cases of MSA. Scale bar = 10 μm.

from 6 nm to 15 nm, suggesting two populations with overlapping diameter measurements. A bimodal distribution was not apparent for fibrils obtained after 2 cycles or 4 cycles, which had a smaller mean diameter (Supplementary Fig. 7). Amplified MSA fibrils and $IV_{Tris}$ fibrils generally had a broad distribution of diameter measurement while control amplified fibrils had a narrower distribution (Supplementary Fig. 8). Limited precision of diameter measurements and variable rotational orientation of fibrils on the EM grids may contribute to the variability of diameter measurements among different populations. Negative stain EM images showed that LBD amplified fibrils were primarily associated with amorphous tissue-derived material after 2 and 4 cycles. After 6 cycles, tissue components were rare and frequent fibril-fibril association was observed, in both parallel and crossed orientations.

SDS-PAGE analysis of amplified fibrils indicated that proteolytic cleavage of Asyn occurred during the first four cycles of incubation,

producing fragments with approximately 2-3 kD reductions in size (Supplementary Fig. 9). Western blot analysis demonstrated that these fragments were produced by cleavage at the C-terminus, likely by proteases present in insoluble fractions from brain tissue (Supplementary Fig. 10). This C-terminal cleavage of Asyn that occurs during fibril amplification corresponds to a high percentage of C-terminal cleavages present in Asyn fibrils extracted from human brain tissue[48–50]. These C-terminal 30 amino acids are disordered and not part of the beta sheet structure in Asyn fibrils isolated from LBD and MSA postmortem tissue[37,38]. The relative levels of truncated protein were reduced substantially after 6 amplification cycles. In addition, levels of tissue-derived proteins were also substantially reduced relative to Asyn (Supplementary Fig. 9). Analysis of Asyn fibrils by proteinase K digestion showed distinct band patterns for $IV_{Tris}$ fibrils compared to amplified fibrils, but relatively small differences in band patterns between different types of amplified fibrils. Proteinase K produced fragments with relative molecular weights predominantly in the range of 10 to 15 kD that were similar between LBD and MSA fibrils, which may be explained by the fact that these small cleavages of 1–5 kD occur in the disordered N- and C-terminal regions. However, a faint band at approximately 7 kD was visible in PK digests of MSA fibrils but not LBD fibrils.

## 2D classification of single-particle cryo-EM data indicates a combination of single protofilament and two-protofilament fibrils with very low twist

We collected single-particle cryo-EM data on a sample of the isotopically-labeled LBD fibrils used for SSNMR analysis. Images from several 2D class averages obtained after 2D classification in RELION are consistent with a two-protofilament structure with pseudo-$2_1$ helical screw symmetry (Fig. 3a, b). We also observed 2D classes that appeared consistent with single protofilament fibril structure, based on comparison of the diameter and density pattern to two protofilament classes (Fig. 3c, d). This observation indicates that the amplified fibrils comprise a combination of single protofilament and two protofilament fibrils, which is consistent with the bimodal distribution observed in diameter measurements from negative stain TEM images (Supplementary Fig. 7). All 2D classes displayed very low twist, which limits the ability to obtain a high-resolution 3D cryo-EM density map with helical reconstruction[51–54].

## Solid-state NMR spectroscopy of LBD fibrils

We leveraged the templated fibril amplification methodology to produce uniformly $^{13}C,^{15}N$ and $^{13}C,^{2}H,^{15}N$ labeled (Fig. 4a, b) and [2-$^{13}C$]-glycerol, uniform-$^{15}N$ labeled fibrils from the LBD1 case. Six cycles of amplification were utilized to generate sufficient material for the SSNMR studies. The SSNMR spectra from these multiple independently prepared amplified fibril samples indicated a high degree of consistency in the amplification protocol. The $^{13}C$-$^{13}C$ 2D correlation spectrum displays a set of sharp peaks that can be individually assigned to resonances within the core of the fibril (residues ~34 to 46 and 62 to 96) (Fig. 4a). This spectrum suggests a highly ordered fibril core and a disordered N- and C- terminus, consistent with previous studies of fibrils and of Asyn in particular[55]. We extended the resonance assignment of the non-amyloid beta component of plaque (NAC) core using uniformly-$^{13}C$, $^{2}H$, $^{15}N$-labeled (uCDN) fibrils, prepared from uCDN Asyn monomers which were then amplified from tissue in 100% $^{1}H_2O$ ($H_2O$) buffer, allowing for the protonation of exchangeable sites uniformly through the fibril. After amplification, fibrils were repeatedly washed with 100% $^{2}H_2O$ ($D_2O$) buffer. This procedure produces fibrils

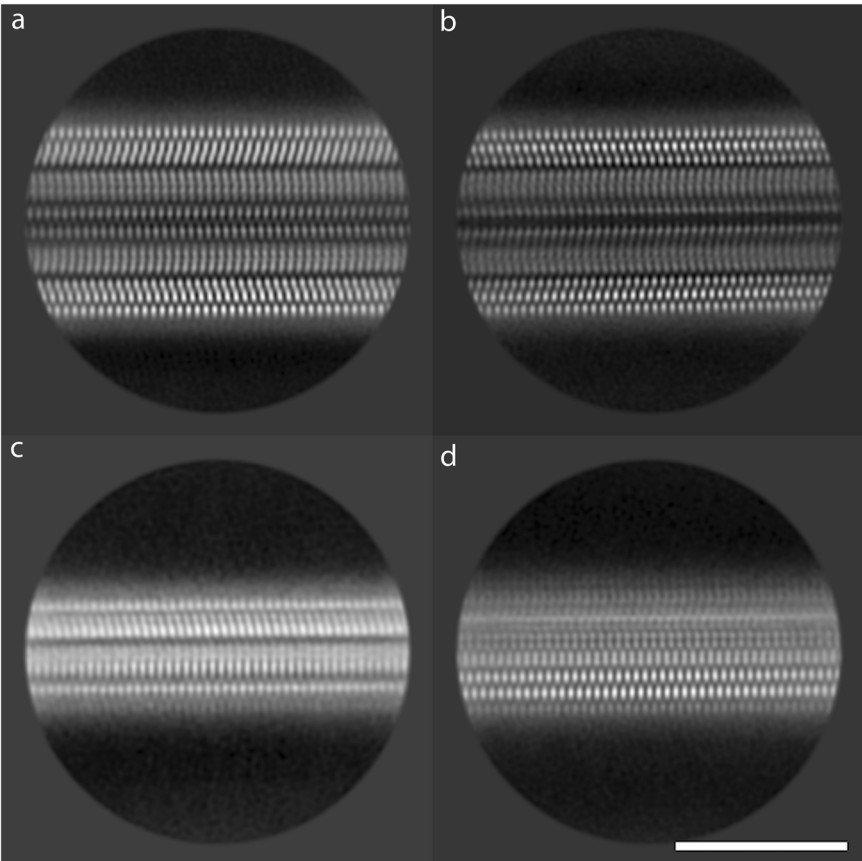

**Fig. 3 | Cryo-EM 2D class averages for LBD amplified fibrils. a, b** Examples of 2D class averages for two protofilament fibril classes obtained from single particle cryo-EM data. **c, d** Examples of 2D class averages for single protofilament fibril classes obtained from single particle cryo-EM data. Scale bar = 10 nM.

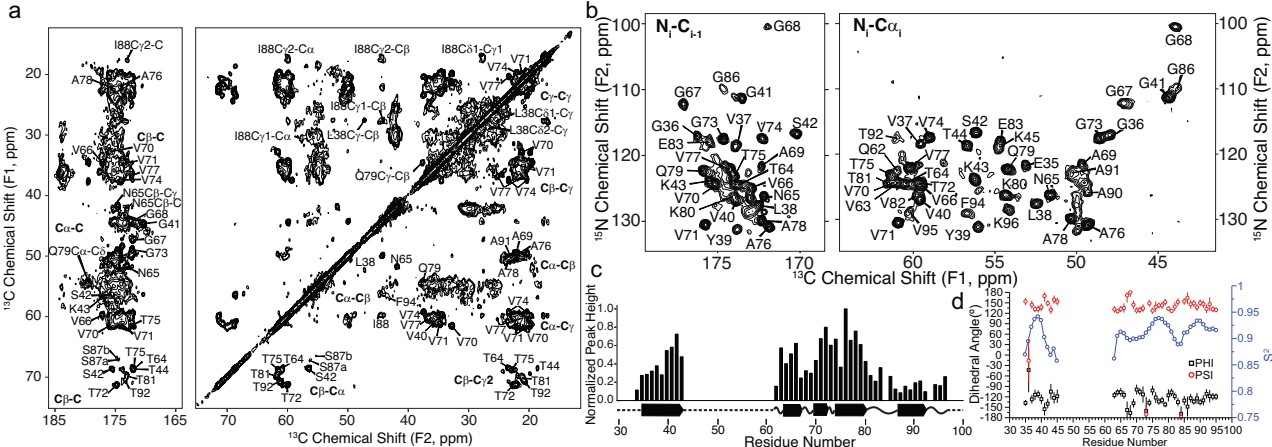

**Fig. 4 | SSNMR assignments, long range contacts, and MPL measurements showing 2 protofilaments are the dominant population of fibrils. a** $^{13}C$-$^{13}C$ correlation of $^{13}C$-$^{15}N$ labeled LBD fibril showing carbonyl (left panel) and aliphatic (right panel) with unambiguous assignments labeled on the spectrum. Data was acquired at 750 MHz $^1H$ frequency with 16.667 kHz magic-angle spinning and sample temperature 10 ± 5 °C with 75 ms DARR mixing. **b** 2D projections of a (left) CO-N-H 3D and (right) CA-N-H experiment with backbone assignments of a $^{13}C$, $^2H$, $^{15}N$ labeled sample. Data in (**b**) were acquired at 750 MHz $^1H$ frequency with 33.333 kHz magic-angle spinning, using a 6 ms $^{15}N$-$^{13}C$ cross polarization. 3Ds were collected using non-uniform sampling followed by reconstruction using SMILE prior to Fourier transformation. **c** Peak heights are plotted for unambiguous assignments from a uniformly sampled CANH 3D spectrum collected on the uniformly $^{13}C$,$^{15}N$-labeled sample. Bars are normalized to the peak height for residue A76. Predicted secondary structure from TALOS-N is plotted below the peak height plot for the LBD tissue-seeded sample. **d** TALOS-N derived dihedral Phi and Psi angles reveal a beta-sheet structure. Predicted RCI chemical shift ordered parameter is plotted in blue showing highly ordered ($R^2 > 0.9$) regions between G36 and G41, T64 and T81, and A85 to V95.

that remain highly protonated at the amide of the backbone of the fibril core but are perdeuterated in the N- and C- termini, making these resonances invisible to $^1H$ detection experiments. In contrast, the backbone amide $^1H$ atoms within the fibril core are protected from chemical exchange with the $D_2O$ and so the amide SSNMR signals are well resolved. Thus, fibrillization in $H_2O$ and washing with $D_2O$ has the effect of increasing the resolution and sensitivity of the core residues for assignment and structural experiments. Figure 4b shows a representative 2D projection of a (Fig. 4b, left) CO-N-H 3D and (Fig. 4b, right) CA-N-H experiments with the backbone assignments of $^{13}C$, $^2H$, $^{15}N$ indicated, with the $^1H$-$^{15}N$ 3D projection and per residue linewidths shown in Supplementary Fig. 11. In addition to confirming the backbone $^{13}C$ and $^{15}N$ assignments determined for the uCN sample, additional assignments were made for residues K43, Q62, E83, G84, A85, and G86, which were better resolved in the $^1H$-detected experiments than in the $^{13}C$-detected versions. We attribute this to the higher sensitivity and improved relaxation properties of these regions of the spectra obtained from the uCDN sample. We estimated the relative protonation quantity and the rigidity of the backbone resonances by measuring the intensity of the cross peak of the 3D CA-N-H data (Fig. 4c). The data provided by the $^{13}C$ and $^1H$ detection experiments were leveraged to yield dihedral restraints (Fig. 4d) and predicted random coil index order parameter (RCI S2), revealing LBD fibrils form highly ordered beta-sheet fibril core structure involving E34 to K45 and V63 to V95. Notably, there is a high correlation between the RCI order parameter and the measured intensity pattern of the 3D CA-N-H peaks (Fig. 4c). Additional details regarding SSNMR pulse sequences, spectra and assignments are outlined in Supplementary Notes 1, 2, Supplementary Figs. 11–20 and Supplementary Tables 9–12.

We further measured long-range $^{13}C$-$^{13}C$ distances based on correlations from 2D and 3D spectra in order to generate a structural model of the LBD amplified fibrils. Figure 5a shows the assignment strips of long-range unambiguous correlations between regions of the fibril that may be assigned using a $^{13}C$-$^{13}C$ correlation experiment with 12 ms of PAR mixing (see also Supplementary Fig. 13). At this mixing time, the PAR experiment reports on distances up to approximately 8 Å, including a number of correlations between residues that are far apart in primary structure but must be proximate in the 3D fold. For example,

L38CA shows correlations to A78CA/CB; V70CB shows correlations to N65CA/CB; G73CA shows a key correlation to F94CD and F49CB. The 2D also reveals cross peaks from S87CB to either A78CA or A91CA, as well as multiple cross peaks between the I88 methyl groups and the Q79 and V77 spin systems. To improve the resolution of the spectra further, we performed a 3D experiment including two different types of mixing (Supplementary Fig. 14). The first, RFDR, transfers polarization from one $^{13}C$ to other $^{13}C$ sites in the same residue; the second, PAR, then transfers the magnetization over longer ranges to derive structurally useful distances. The 3D data set provides several additional sets of peaks to validate and extend the interpretation of the 2D. For example, T92CB and T72CB are clearly resolved in the 3D $^{13}C$-$^{13}C$-$^{13}C$ and independently exhibit distinct cross peaks with F94CD (Fig. 5b). Additional correlations from the 3D $^{13}C$-$^{13}C$-$^{13}C$ include (Supplementary Fig. 15) cross peaks between the G41CA and A69CA/CB and V70CB; N65CB and V70CA/CB/CG; G68CA to S42CA/CB and G41C; T75CB to T92CB; and T92CB to V74C'/CA/CB and G73C'. We additionally used $^1H$ detection methodologies to gather long range correlations. Figure 5c shows contacts identified within a hCAhhNH experiment showing close contact between L38 with A76 and V77 and G41 with A69. The set of $^{13}C$, $^{15}N$ assignments for LBD1 fibrils amplified with uCN Asyn monomer are published in Biomolecular NMR Assignments[56].

**Atomic model of amplified LBD Asyn fibrils**
Mass-per-length (MPL) is an invaluable constraint on the supramolecular organization of amyloid fibrils that can be measured experimentally. Dark-field transmission electron microscopy (TEM) of unstained, amplified LBD Asyn fibrils from the same batch as those used to prepare the uCN SSNMR sample was used to measure an MPL of 60 ± 14 kDa/nm (Fig. 5d). With a molecular weight of the uCN labeled monomer of 15.244 ± 2 kDa as determined by MALDI-TOF MS (Supplementary Fig. 16), this average MPL observation is consistent with two molecules of Asyn per 4.8 Å β-sheet spacing. A combination of two protofilament fibrils and single protofilament fibrils are observed in 2D class averages obtained from single particle cryo-EM data (Fig. 3). To calculate an atomic model for the two protofilament fibrils, we utilized an assumption that the beta strands along the fibril axis within each protofilament are parallel and in-register, and composed of two

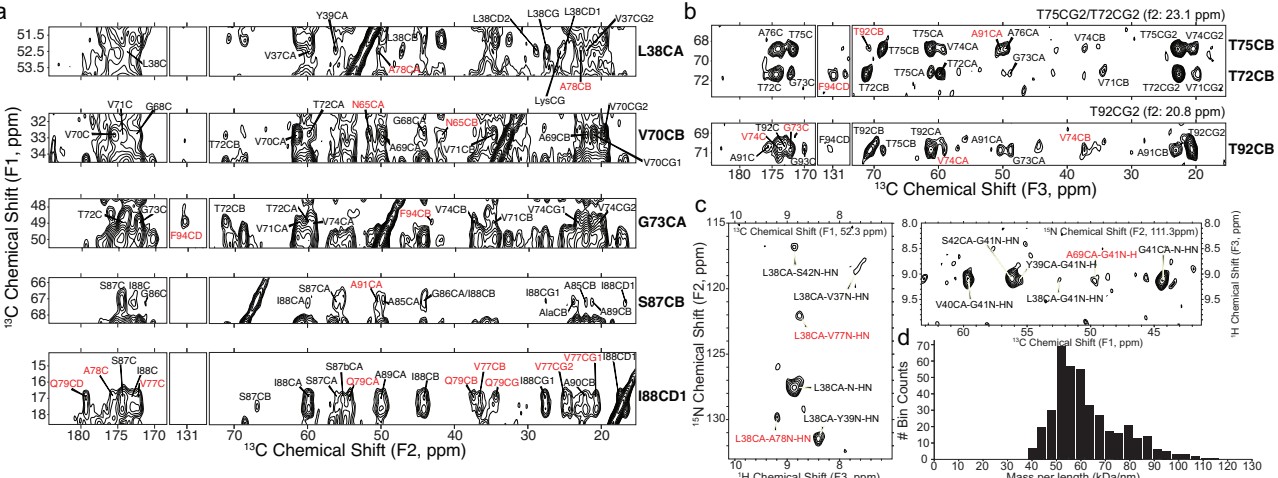

**Fig. 5 | Long-range correlations determined by SSNMR. a** Long-range unambiguous correlations identified using 12 ms of PAR mixing between L38 V70, G73, S87, I88 found uniquely resolved within a ¹³C-¹³C 2D correlation experiment. **b** 2D strip plots from a 3D ¹³C-¹³C-¹³C correlation spectrum showing correlations from three threonine (T72, T75, T92) residues that are disambiguated in F1 and F2 relative to the 2D spectrum. **c** Long-range contacts identified within a hCAhhNH experiment showing close contact between L38 with A76 and V77 and G41 with A69. **d** Mass per unit length histogram of the LBD fibril measured with dark-field, unstained TB-TEM micrograph.

symmetric protofilaments, which is consistent with the observation of a single set of peaks in the SSNMR spectra. Furthermore, the single set of peaks in the NMR spectra indicates that the single protofilament fibrils have the same fold as two protofilament fibrils.

In the calculation of the structure of amplified LBD Asyn fibrils, we utilized the PASD protocol[57,58] to perform the automated assignment of long-range correlations for the generation of distance restraints in Xplor-NIH[59]. The strict symmetry facility within Xplor-NIH was used to simplify the calculation by representing the fibril using a single protomer subunit, replicated by rotational and translational symmetry to form a 10-subunit fibril in which a pair of five subunit two-fold symmetric protofilaments interact laterally. Violation statistics for high-likelihood restraints in the context of the calculated structure bundles at each stage of PASD are summarized in Supplementary Fig. 18a, b. A summary of simulated annealing protocols in the PASD algorithm is given in Supplementary Table 11. Given the resonance list for the ¹³C-detected nuclei, the initial PASD assignment process resulted in 1099 cross-peaks from data collected on the uCN and diluted samples for which only non-short-range (between residues separated by 3 or more residues in primary sequence) assignments were possible. $262 \pm 7$, or 24% of these peaks contained assignments which were satisfied by the final ensemble of 11 structures. This initial procedure resulted in an average ambiguity of $20 \pm 22$ peak assignments per peak. Following the network analysis stage where reinforcing assignments are given higher weight and including a filter to remove any peaks with a protomer ambiguity greater than two, the fraction of peaks satisfying the final bundle was at 31%, and the overall ambiguity dropped to $3 \pm 2$. Three rounds of PASD structure calculations (pass 2, pass 3, and pass 4) simultaneously removed nearly all the violating restraints and gradually added back the satisfied restraints. A hybrid manual/PASD pass5 structure calculation-based refinement was then performed, where the original peaks were reassigned based on structures generated using the pass4 peak lists, increased the number of satisfied restraints to $183 \pm 2$.

The structural model in Fig. 6a is a cartoon representation of the lowest energy structure produced from the series of calculations of two protofilaments, showing residues A30–N103. The 10 lowest energy structures of the LBD fibril are represented in Fig. 6b and the restraints used for structure calculations are presented in Supplementary Fig. 19. Structural features include a tight interface between the two protofilaments, a core composed exclusively of hydrophobic and polar

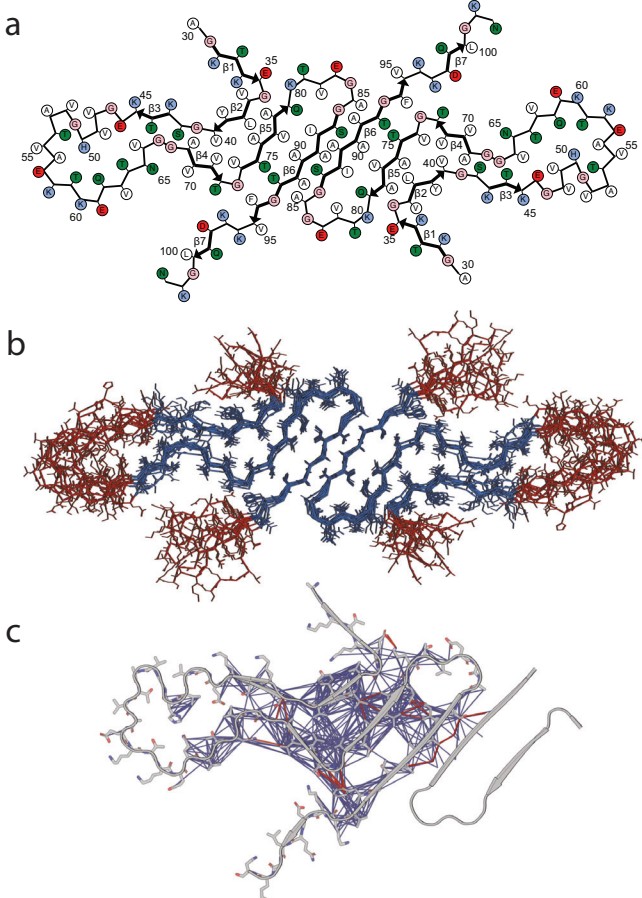

**Fig. 6 | Structure of LBD amplified fibrils. a** Cartoon representation of the LBD fibrils determined via SSNMR studies. **b** 10 lowest energy structures of LBD fibril determined. Blue represent well-ordered regions with large amount of intermolecular contacts; red represent relatively disordered regions with few intermolecular contacts detectable using CP based polarization techniques. **c** Inter-residue contacts identified from SSNMR CP based experiments. Magenta lines are intramolecular contacts manually identified within spectra, blue lines are correlations identified utilizing the PASD algorithm for probabilistic assignments.

residues, and charged residues lining the periphery. Notably, the residues that make up the hydrophobic core of the fibril (V71 to V82) have been described before as essential to fibril formation[60]. Additionally, a likely salt bridge between E35 and K80 was not observed directly but was added as a modeling restraint during the final refinement calculation based on a combination of the FS-REDOR result (measuring the $^{15}N$-$^{13}C$ distance across the salt bridge) and the observation that these two residues were consistently in close proximity in the converged structures, even without the salt bridge restraint. The top ten structures ranked according to lowest total energy in Xplor-NIH are overlaid by all heavy atoms in Fig. 6b. Residues 35–46 and 62–96 of these structures converged to a backbone RMSD of 1.2 Å and a heavy atom RMSD of 1.7 Å, a resolution that allows many individual side chains to be resolved (Supplementary Table 12). The ten lowest energy structural models have been deposited in the Protein Data Bank (PDB) under accession number 8FPT.

We observe several notable structural features of the final determined structure. L38 forms a steric zipper with A76, V77, and A78 (Fig. 6a). G41 and S42 pack tightly against G68 and A69 (Fig. 6a). Correlations between the F94 aromatic carbons with T72 and G73, as well as the strong cross peaks between V71 with V74 and T75 with T92 are demonstrated. The interface between the protofilaments was particularly notable but also a challenge to assign manually because several short- and medium-range correlations are present (Fig. 6c). The PASD calculations result in convergence to the anti-parallel inter-filament beta-strand conformation. The resulting structure clarifies that S87 and A78 are part of the strong network of peaks between I88 with V77 and Q79, and that additional correlations from S87 to A91 on the other protofilament. We utilized an unrestrained molecular dynamics (MD) simulation to analyze the thermodynamic stability of the structural model determined by SSNMR and found that the core residues are qualitatively unchanged after 200 ns of equilibrium MD

(Supplementary Fig. 20). Scripts and molecular dynamics simulation data have been submitted in the DRYAD public repository [https://doi.org/10.5061/dryad.tx95x6b4z]. These SSNMR data and MD simulations demonstrate that the network of interactions among these residues defines a stable anti-parallel filament interface, although it is possible that other beta-strands could also form interfaces, as observed in several in vitro forms of Asyn fibrils which exhibit similar folds but different filament interfaces.

## NMR spectra from additional LBD cases indicate similar structure

We collected SSNMR data from two additional LBD cases, utilizing amplified fibril preparations from cortical samples (middle frontal gyrus) for comparison to the structure of the LBD1 sample from caudate. We utilized lower amounts of isotopically-labeled amplified fibrils from these cases for the purpose of comparing NMR spectra. The signal intensities were significantly lower than that of LBD1, but were sufficient to compare 1D spectra with LBD1 for two additional cases and 2D spectra for one additional case. The positions and relative intensities of peaks in the aliphatic region of the 1D $^{13}C$ spectra between the three samples are highly similar, especially for diagnostic chemical shifts of Thr, Ala, and Val residues, indicating a high degree of structural similarity in their fibril cores (Fig. 7a). Additionally, the similar positions and relative intensities of peaks in 2D spectra for amplified fibrils from cases LBD6 and LBD1 (Fig. 7b, c) further indicate a high degree of homogeneity in their secondary structures.

## SPARK analysis validates key amino acid residues required for LBD fibril growth and stability

We measured growth rates of amplified LBD fibrils in the presence of WT and mutant Asyn monomer, utilizing FlAsH dye to detect close association of two or more bicysteine tagged Asyn monomers upon

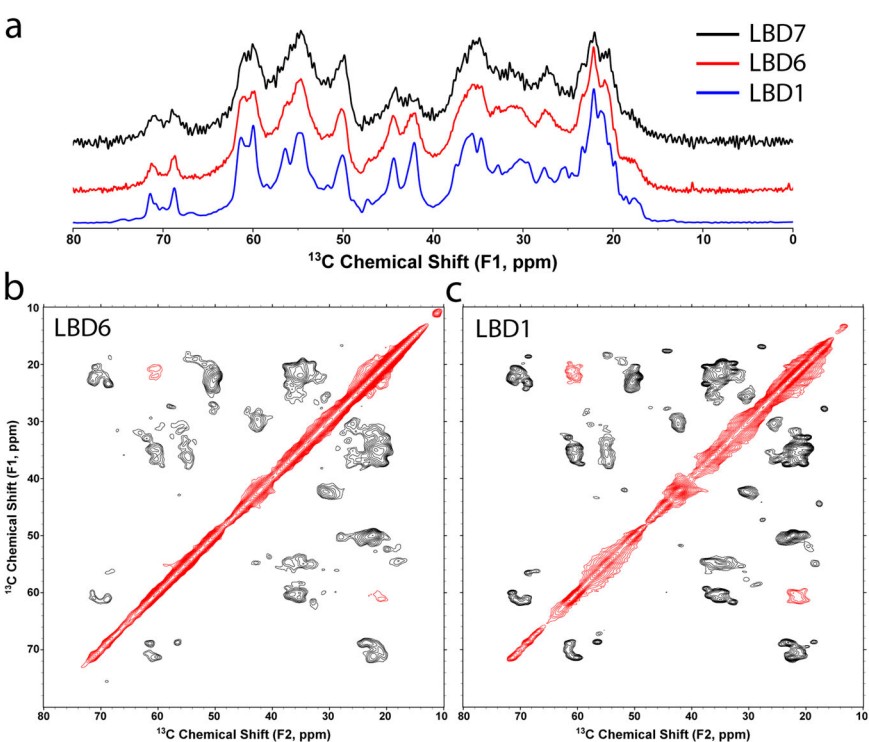

**Fig. 7 | Spectral comparison of amplified fibrils from three LBD cases. a** 1D $^{13}C$ cross polarization spectra show the rigid fibril cores of LBD1 (blue), LBD6 (red) and LBD7 (black) amplified fibrils. The aliphatic region of the spectra is shown, highlighting the similarities in the peak positions and relative intensities between the samples. **b, c** 2D $^{13}C$-$^{13}C$ correlation spectra with 1 ms POSTC7 mixing of (**b**) LBD6 and (**c**) LBD1 fibrils. Black contours (positive) correspond to correlations between bound $^{13}C$ nuclei pairs, and red contours (negative) correspond to correlations of $^{13}C$ nuclei two atoms apart. The chemical shifts are highly similar overall, in particular for diagnostic sidechains of Ala, Thr and Val residues.

incorporation into fibrils[39]. In this approach, which we refer to as Synuclein Polymorph Analysis by Relative Kinetics (SPARK), we examine relative differences in fibril growth rates that distinguish different fibril forms. We initially compared the effect of familial PD-associated Asyn mutations on growth rates for in vitro assembled (Tris) fibrils, LBD amplified fibrils and MSA amplified fibrils (Fig. 8a). For in vitro fibrils, we observed very inefficient growth, relative to WT, with the A53T and E46K mutations, but no effect on growth for H50Q[39]. In contrast, growth rates for LBD fibrils are similar for this set of three mutant monomers relative to WT monomer. Growth rates for MSA amplified fibrils were variable among the four cases, limiting statistical

analysis, but appear to be increased for A53T and reduced for E46K, which is consistent with a previous study demonstrating that MSA brain tissue extracts cannot seed aggregation of E46K Asyn in 293 cells[61].

Based on the SSNMR structural model of LBD fibrils, we identified several residues central to fibril structure and examined the effect of these mutations on fibril growth. The S87K mutation dramatically reduces fibril growth for all three polymorphs (Fig. 8b). In contrast, a conservative S87Q mutation has little effect on in vitro and MSA fibrils, but substantially reduces LBD fibril growth. Similarly, an A76T mutation greatly inhibits LBD fibril growth, but has minimal effects on in vitro and MSA fibrils. Finally, G68Q reduces the growth of all three polymorphs. Thus, we identified four mutations that significantly reduce LBD fibril growth as predicted by the structural model, and found that two mutations (S87Q and A76T) appear uniquely important for LBD fibril growth compared to MSA and $IV_{Tris}$ fibrils.

## Comparison of structural features among Asyn fibrils

The ordered core region of the amplified LBD Asyn fibrils (PDB 8FPT) is highly similar to the recently reported cryo-EM structure of fibrils extracted from postmortem PD and LBD brain tissue (Fig. 9) (PDB 8A9L[38]). Strands comprising residues 84-95 and 67-81 in the SSNMR and cryo-EM models overlay with root-mean-square deviation (r.m.s.d) values of the main-chain atoms of 0.73 Å and 1.0 Å, respectively. Moreover, these residue ranges have comparable numbers of satisfied SSNMR distance restraints (Fig. 9d, e). In the 'hinge' region (residues 47-61), there are very few SSNMR assignments and restraints. Likewise, the cryo-EM model reported higher B-factors in this same region. The N-terminal strand comprising residues 36–45 is more divergent between the two models, with a higher r.m.s.d. value (3.0 Å). Both models contain proteinaceous entities packed against the predominantly hydrophobic residues 85–93. In the SSNMR two-protofilament model, this is the unique interface formed between two protofilaments with pseudo-$2_1$ helical screw symmetry. For the cryo-EM model, which comprises a single protofilament, this is the unidentified peptide termed island B. The cryo-EM model also contains an additional unidentified island, a peptide packed against residues 50–55. In the SSNMR model, this residue range contains too few assignments and restraints to ascertain potential peptide interactions. The fold of LBD fibrils differs substantially from MSA and in vitro assembled Asyn fibrils. However, one consistent feature shared by LBD, MSA, and in vitro conformers is the presence of an L-shaped motif formed by hydrophobic residues 69-79 (Supplementary Fig. 21), which is consistently buried between two other hydrophobic strands to form a three-layered structure.

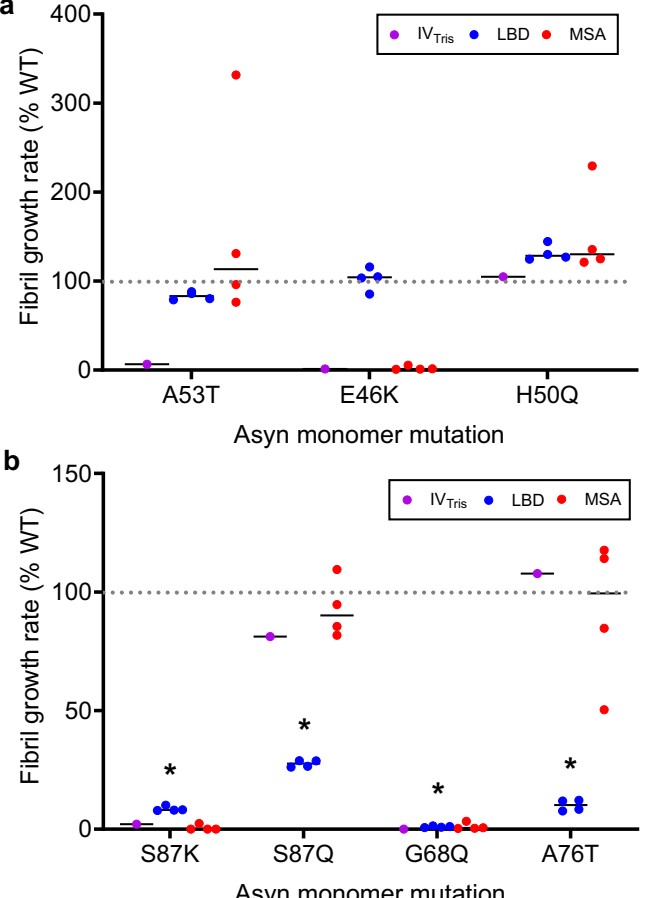

**Fig. 8 | Fibril growth rates for in vitro assembled fibrils and amplified fibrils in the presence of mutant Asyn monomer. a** WT-normalized growth rates in the presence of Asyn monomer with familial PD mutations A53T, E46K and H50Q. **b** WT-normalized growth rates in the presence of Asyn monomer with S87K, S87Q, G68Q and A76T mutations. Each data point represents the mean growth rate ($n = 3$ replicates) obtained for amplified fibrils prepared from each individual autopsy case for LBD (blue) and MSA (red), in addition to $IV_{Tris}$ fibrils (magenta). Fibril growth rate measurements were performed for amplified fibrils from four LBD cases and four MSA cases. Asterisks indicate the mutant monomers with statistically significantly lower growth rates relative to WT monomer for LBD cases. Growth rates for MSA amplified fibrils had higher variability, including for WT monomer, and differences between mutant and WT monomer were not significant. The results indicate that four mutations (S87K ($p = 0.009$), S87Q ($p = 0.008$), G68Q ($p = 0.007$) and A76T ($p = 0.009$)) significantly reduce growth rates of LBD fibrils. Two of these mutations (S87K and G68Q) also appear to reduce $IV_{Tris}$ and MSA fibril growth rates, but the other two (S87Q and A76T) do not. Data were analyzed with the unpaired, two-tailed t-test (Welch) and corrected for multiple comparisons with the Holm-Bonferroni method (significance level of 0.05). Data presented as mean ± SD, independent experiments = 2. Similar results were obtained from two independent experiments. Source data are provided as a Source Data file.

## Discussion

We developed a method to amplify Asyn fibrils from LBD postmortem tissue, enabling high-resolution analysis of fibril structure. 2D classification results from single-particle cryo-EM indicate a combination of single protofilament and two protofilament fibrils with very low twist. Several hundred SSNMR distance and dihedral restraints were used to develop a structural model based on backbone dihedral angle determination with TALOS-N[62] and Xplor-NIH[59] simulated annealing calculations. The fold of LBD amplified fibril is highly similar to the fold in the single protofilament structural model derived from single-particle cryo-EM studies of LBD fibrils with high twist. These results substantially extend the structural characterization of LBD Asyn fibrils by providing insight into the structure of fibrils with low twist that have not been amenable to high-resolution structure analysis with cryo-EM. Furthermore, by facilitating the production of large quantities of Asyn fibrils with the LBD fold, this amplification method expands approaches for studying disease mechanisms, imaging agents, and therapeutics targeting Asyn.

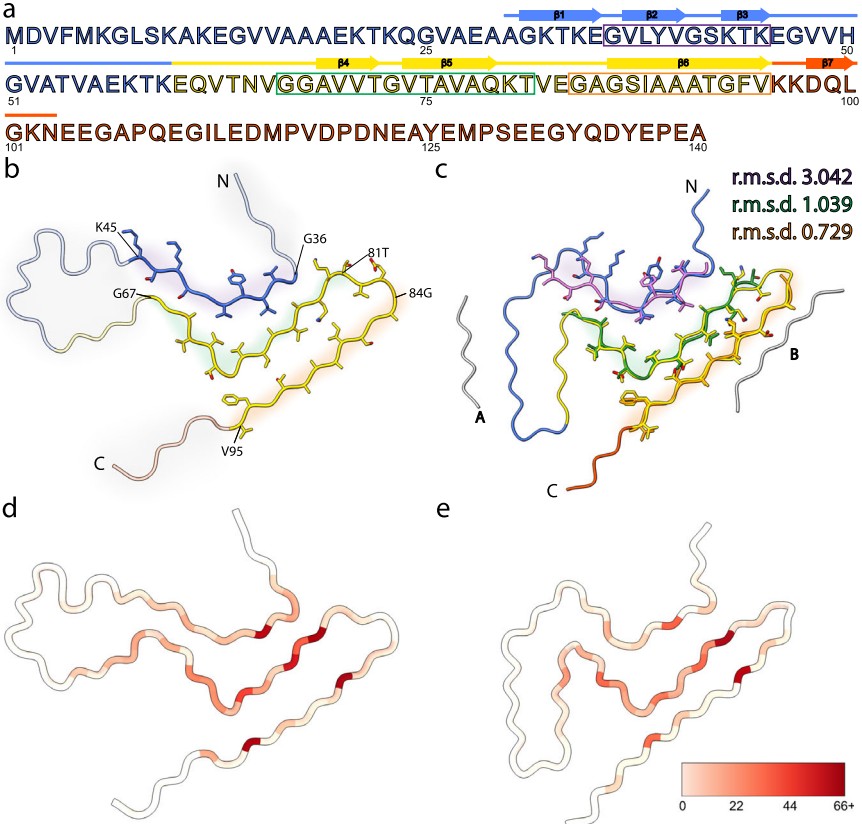

**Fig. 9 | SSNMR structure of amplified Asyn resembles cryo-EM structure of extracted filaments. a** Primary sequence of human Asyn, with beta-strands annotated with arrows. The amphipathic N-terminal region, hydrophobic non-amyloid beta-component of plaque (NAC) region, and acidic C-terminal region are colored blue, yellow, and red, respectively. The purple, green, and orange rectangles denote the residue ranges used for subsequent comparison studies. **b** Atomic model of SSNMR LBD Asyn core structure with labeled N- and C- termini (PDB 8FPT). Areas of low confidence are depicted with transparent cartoons with grey highlights. Residue ranges modeled with high confidence are depicted as cartoons with sidechains as sticks and are highlighted in purple, green, and orange. **c** Atomic model of cryo-EM LBD Asyn core structure (PDB 8A9L), colored as in (**b**). The unidentified proteinaceous islands are in grey. High-confident residue ranges of the SSNMR structure are overlayed in purple, green, and orange, and root-mean-square deviations (r.m.s.d) of the main-chain atoms were calculated. **d** Number of satisfied NMR-derived interatomic distances mapped onto the lowest-energy structure using Xplor-NIH calculations on a per-residue basis using initial assignments from PASD. The color scale extends from white, indicating few or no NMR-derived distances were observed for the residue, to red, indicating many NMR-derived distances were observed for the residue. **e** Number of satisfied NMR-derived distances mapped onto the cryo-EM Asyn core structure (PDB 8A9L).

There are several possible explanations for the identification of a combination of single and two protofilament fibrils in this study and only single protofilament fibrils in the recent cryo-EM study[38]. In 2D classification of single-particle cryo-EM data for amplified fibrils, we observe classes of single protofilament fibrils in combination with the two-protofilament classes. The amplification protocol utilizes fibril seeds obtained in a 1% Triton insoluble fraction, which is significantly different from the insoluble fraction obtained after extraction in 2% sarkosyl for the reported cryo-EM study. In contrast to extraction protocols utilizing sarkosyl, an ionic detergent with high solubilizing strength, we do not observe fibrils that are clearly isolated from a large amount of amorphous tissue-derived material after extraction in Triton, a weaker nonionic detergent. Thus, we were not able to directly assess diameter and structure of the fibril seeds used for amplification. It is possible that 2% sarkosyl selectively extracts single protofilament fibrils or converts two protofilament fibrils to single protofilament fibrils, although we did not observe a change in diameter after treatment of amplified fibrils with 2% sarkosyl. Quantification of fibril diameter in negative stain TEM images indicates an increase in fibril diameter with a bimodal distribution in later cycles, suggesting that the proportion of two protofilament fibrils increases with additional cycles of amplification. It is possible that the amplification protocol favors the accumulation of two protofilament fibrils over single protofilament fibrils, based on faster growth rates and/or higher fragmentation rates for two

protofilament fibrils. It is also possible that the amplification protocol promotes the conversion of single protofilament fibrils to two protofilament fibrils by promoting fibril-fibril association events, particularly in later cycles of amplification that obtain relatively pure fibril preparations with minimal tissue-derived material. In situ studies of Asyn fibril structure in postmortem tissue, although technically challenging, could provide more definitive information on the presence and relative abundance of the two classes of fibrils.

By substantially expanding the availability of LBD fibril quantities beyond what can be obtained from frozen postmortem tissue samples, the amplified fibril preparations reported here provide avenues for studies of disease mechanisms and therapeutics. For example, LBD amplified fibrils facilitate studies of Asyn fibril growth, as we demonstrate in the SPARK analysis. Amplified fibrils can also be used to seed LBD Asyn fibril accumulation in cell culture models and animal models, where the LBD fold is likely to be maintained in accumulating fibrils, a prediction that can ultimately be verified by structure analysis of fibrils extracted from model systems. The use of amplified fibrils in vitro or in model systems can facilitate the development of imaging agents by enabling in vitro and in vivo measurements of binding properties for Asyn ligands. Finally, studies in vitro and in model systems can facilitate the development of candidate therapeutics capable of inhibiting fibril accumulation. Our LBD fibril amplification method is based in part on previous studies utilizing seeded fibril growth for SSNMR

studies of amyloid beta fibrils isolated from Alzheimer's disease brain tissue[63,64]. Other Asyn fibril amplification methods, utilizing tissue extracts or cerebrospinal fluid as starting material, have produced Asyn fibrils with structures that diverge from the structures determined for tissue-derived fibrils in MSA and LBD, indicating that some aspects of the amplification protocol described here, such as fragmentation and incubation conditions, are important for templating structure during amplification[65–67].

The fold of LBD Asyn fibrils is distinct from MSA Asyn fibrils and from all in vitro assembled polymorphs reported to date. LBD fibril structure appears to be highly similar among LBD cases, with similarity in the cryo-EM density maps obtained for six LBD cases in the study by Yang et al.[38] We have observed a high degree of similarity in 2D SSNMR spectra for one additional LBD case, and have also observed highly similar 1D spectra for a second additional LBD case. The SSNMR structural model reported here is derived from the amplification of LBD fibrils from caudate, demonstrating a high degree of similarity to the subset of twisted fibrils isolated from the cortex in the cryo-EM study[38]. However, further studies are needed to determine whether subtle structural variations exist among LBD cases and whether structural heterogeneity occurs within individual cases, including across different brain regions. Structural variations, if present, may relate to the spectrum of clinical features observed in LBD, such as the variable timing of dementia onset. They may also relate to genetic and environmental risk factors identified for PD/DLB. For example, different genetic and environmental factors may give rise to differences in metabolites, post-translational modifications, and interacting proteins that drive variations on a set of core structural elements.

SPARK analysis validates and extends the structure analysis for LBD fibrils. Using SSNMR structural data, we identified four amino acid residues likely to be central to fibril structure, and validated that these residues are critical for LBD fibril growth and stability. Two of the amino acid residues appear uniquely important for LBD fibrils when compared to in vitro and MSA amplified fibrils, providing additional insight into the unique structural features of LBD fibrils. These differences in fibril growth rates likely reflect structural constraints specific to each fibril form, where a mutation of a specific amino acid residue has the potential to alter kinetics based on interactions with neighboring beta strands within a single monomeric unit or alternatively at the protofilament interface. Analysis of additional residues could extend this kinetic characterization of fibril structure, including residues predicted to be important for the protofilament interface and pseudo-helical screw symmetry, which has also been observed in other amyloid fibrils including tau fibrils from AD tissue[68,69]. SPARK analysis could also be utilized to probe the significance of structural heterogeneity, if identified within individual LBD autopsy cases, or to analyze the significance of structural variations if observed across different LBD cases.

Further investigation of Asyn fibril structure in LBD can include analysis of fibrils obtained with additional extraction and amplification methods, with the goal of improved understanding of the conditions that give rise to single protofilament and two protofilament fibril forms, as well as twisted fibrils. Understanding conditions responsible for generating structural features specific to LBD fibrils may provide insight into disease mechanisms in LBD. Joint refinement of structural models based on SSNMR and cryo-EM data could enhance the structural information available to guide the development of imaging ligands and therapeutics, which will also be aided by the analysis of additional LBD cases.

## Methods
### Ethical statement - demographics and clinical information of participants
The Movement Disorders Brain Bank, Washington University, St. Louis, MO, provided clinically and neuropathologically well-characterized postmortem frozen brain tissue (Supplementary Table 8). The Human Research Protection Office at Washington University in Saint Louis approved this study. Written informed consent to perform a brain autopsy was obtained from all participants. After death, the immediate next-of-kin were contacted and confirmed consent for brain removal and retention of brain tissue for research purposes. LBD1 was selected and amplified for SSNMR structural characterization.

### Asyn monomer production (natural abundance)
Natural abundance recombinant Asyn protein was prepared using *E.Coli*. BL21(DE3)RIL bacterial cells were transformed with the pRK172 plasmid containing the WT Asyn construct[39]. Transformed colonies were selected and inoculated in 2 L flask containing 250 ml sterilized TB (1.2% bactotryptone, 2.4% yeast extract, 0.4% glycerol, 0.17 M $KH_2PO_4$, 0.72 M $K_2HPO_4$) with 50 µg/ml ampicillin, and incubated overnight at 37 °C with shaking. Cultures grown overnight were centrifuged (3900×g for 10 min at 25 °C) and pellets resuspended in 20 ml osmotic shock buffer (30 mM Tris·HCl, 2 mM EDTA, 40% Sucrose, pH 7.2). After 10 min of incubation, cell suspension was centrifuged at 8000×g for 10 min at 25 °C. The supernatant was removed and pellet resuspended in 22.5 ml cold $H_2O$ before adding 9.4 µl 2 M $MgCl_2$ to each tube. After 3 min incubation on ice, suspension was centrifuged at 20,000×g for 15 min at 4 °C. To a fresh tube containing supernatant, streptomycin was added to a final concentration of 10 mg/ml and again centrifuged at 20,000×g for 15 min at 4 °C. To this supernatant, 1 mM dithiothreitol (DTT) and 20 mM Tris·HCl pH 8.0 were added, before boiling for 10 min to precipitate heat-sensitive proteins, which were pelleted at 20,000×g for 15 min at 4 °C. The supernatant was collected and filtered through a 0.45 µm surfactant-free cellulose acetate filter (Corning) before loading onto a 1 ml DEAE Sepharose column (Catalog DCL6B100-50ML, Sigma) equilibrated in 20 mM Tris·HCl pH 8.0, 1 mM EDTA, and 1 mM DTT. The DEAE column was washed with 20 mM Tris·HCl pH 8.0, 1 mM EDTA, 1 mM DTT before eluting Asyn protein in 20 mM Tris·HCl pH 8.0 buffer with 1 mM EDTA, 1 mM DTT and 0.3 M NaCl. Purified Asyn protein was dialyzed overnight in 10 mM Tris·HCl pH 7.6, 50 mM NaCl, 1 mM DTT and stored at −80 °C until use.

For use in amplified fibril production, the purified monomer was filtered through Amicon ultra-4, 50k (Catalog UFC805024, Millipore) cutoff filter to remove any preformed Asyn aggregates. Approximately 70% of Asyn protein was recovered after filtration through the 50 k Amicon Ultra-4. The 50k-filtered Asyn monomer was stored at 4 °C until use in the amplification process. Micro-BCA assay was used to determine the protein concentration of the Asyn monomer.

### Preparation of insoluble fraction seeds from LBD, MSA, and control postmortem tissue
The protocol to sequentially extract human postmortem brain tissue was adapted from our previous publication[70]. Briefly, gray matter dissected from tissue was sequentially homogenized in four buffers (3 ml/g wet weight of tissue) using Kimble Chase Konte™ Dounce tissue grinders (KT885300-0002). In the first step, 300 mg of dissected grey matter tissue was homogenized using 20 strokes of Pestle A in High Salt (HS) buffer (50 mM Tris·HCl pH 7.5, 750 mM NaCl, 5 mM EDTA plus 0.1% (v/v) Sigma P2714 Protease Inhibitor (PI) cocktail). The homogenate was centrifuged at 100,000×g for 20 min at 4 °C and the pellet was homogenized in the next buffer using 20 strokes of Pestle B. Extractions using Pestle B were performed in HS buffer with 1% Triton X-100 with PI, then HS buffer with 1% Triton X-100 and 1 M sucrose, and with 50 mM Tris·HCl, pH 7.4 buffer. In the final centrifugation, the resulting pellet was resuspended in 50 mM Tris·HCl, pH 7.4 buffer (3 ml/g wet weight of tissue). The aliquots of insoluble fraction were stored at −80 °C until use. A similar extraction protocol was followed for LBD, MSA, and control cases.

## Preparation of LBD, MSA, and control postmortem soluble tissue fraction

Dissected gray matter of tissue (100-300 mg) was prepared by homogenization of dissected gray matter in 50 mM Tris, pH 7.4 (3 ml/g wet weight of tissue) using 20 strokes of Pestle A followed by 20 strokes of Pestle B Dounce tissue homogenizers. The aliquots of unfractionated homogenates were stored at −80 °C until use. A 100 µl aliquot of this homogenate was spun down at 100,000 xg and the supernatant was used as the soluble fraction in cell culture experiments.

## Amplification of LBD fibrils from LBD insoluble fraction seeds

We amplified LBD fibrils from gray matter dissected from the caudate brain region. Insoluble fraction (10 µl) containing 3.3 µg wet wt. of tissue was bought to a final volume of 30 µl by addition of 20 mM Tris-HCl, pH 8.0 plus 100 mM NaCl buffer (fibril buffer) in a 1.7 ml microcentrifuge tube. This insoluble fraction was sonicated for 2 min at amplitude 50 in a bath sonicator (Model Q700, Qsonica) with a cup horn (5.5 inch) attachment at 4 °C. To the sonicated seeds, 1.5 µl of 2% Triton X-100 was added to recover the sonicated tissue-derived Asyn fibril seeds. To this mixture, 50k Amicon ultra filtered Asyn monomer was added to a final concentration of 2 mg/ml in a final volume of 100 µl. This mixture underwent quiescent incubation at 37 °C for 3 days, completing the 1st cycle of sonication plus incubation. After the first cycle, the mixture was sonicated at 1 min at amplitude 50, and then an additional 300 µl of 2 mg/ml Asyn monomer was added. The mixture underwent quiescent incubation at 37 °C for 2 days (2nd cycle). Then, sonication for 1 min at amplitude 50 and quiescent incubation for 2 days was repeated for the third cycle, followed by sonication for 1 min at amplitude 50 and quiescent incubation for 3 days for the 4th cycle. After 4 cycles, LBD amplified fibrils were stored at 4 °C until use. The typical scheme of incubations was 3-2-2-3 days for 4 cycles.

Further expansion of the LBD amplified fibrils was performed by centrifuging 60 µl of the 4th cycle LBD amplified fibrils at 21,000 xg for 15 min at 4 °C. The pellet was resuspended in 100 µl of fibril buffer and sonicated for 1 min at amplified 50. To this mixture, Asyn monomer was added to a final concentration of 2 mg/ml in a final volume of 400 µl in fibril buffer. This mix was quiescently incubated at 37 °C for 2 days (5th cycle). After 5 cycles, samples were centrifuged at 21,000 xg for 15 min at 4 °C and the top 300 µl of spent Asyn monomer was moved to a separate tube. The pellet was resuspended by trituration and sonicated for 1 min at amplified 50. After sonication, the previously removed monomer was added back. Next, an additional 2.5 mg/ml of Asyn monomer was added to bring the total volume to 800 µl. This mixture was incubated at 37 °C for 2 days to complete 6 cycles of incubation. The increased monomer concentration (2.5 mg/ml instead of 2 mg/ml) was calculated based on the average decrease in free Asyn monomer due to its incorporation into amplified fibrils. After 6 cycles, fibrils were stored at 4 °C until use. In parallel, insoluble fraction from control tissue samples were also amplified under similar conditions used for LBD amplified fibrils.

## Production of isotopically labeled Asyn monomer

Expression of uniform [$^{13}$C, $^{15}$N] labeled wild-type Asyn was carried out in E. Coli BL21(DE3)/pET28a-AS in modified Studier medium M[71]. The labeling medium contained 3.3 g/L [$^{13}$C]glucose, 3 g/L [$^{15}$N]ammonium chloride, 11 mL/L [$^{13}$C, $^{15}$N]Bioexpress (Cambridge Isotope Laboratories, Inc., Tewksbury, MA), 1 mL/L BME vitamins (Sigma), and 90 µg/mL kanamycin. After a preliminary growth in medium containing natural abundance (NA) isotopes, the cells were transferred to the labeling medium at 37 °C to an OD$_{600}$ of 1.2, at which point the temperature was reduced to 25 °C and protein expression induced with 0.5 mM isopropyl β-D-1-thiogalactopyranoside (IPTG) and grown for 15 h to a final OD$_{600}$ of 4.1 and harvested.

For [2-$^{13}$C]glycerol, uniform $^{15}$N sample diluted to 25% in NA monomer, the labeled monomer was expressed according to the protocol of Tuttle et al.[16] A preliminary culture was grown in one volume of Studier medium M containing 2 g/L ammonium chloride, 2 g/L glycerol, 1 mL/L BME vitamins, and 90 µg/mL kanamycin to an OD$_{600}$ of 2.0. The cells were gently pelleted and resuspended in 0.5 volume of Studier medium M containing 2 g/L [$^{15}$N]ammonium chloride, 4 g/L [2-$^{13}$C]glycerol, 1 g/L [$^{13}$C]sodium carbonate, 1 mL/L BME vitamins, and 90 µg/mL kanamycin. To maximize aeration, the culture was aliquoted as 150 ml per baffled 2-L flask, and grown at 25 °C with shaking at 250 rpm. After an hour the OD$_{600}$ was 4.4, and expression was induced with 0.5 mM IPTG. The culture was harvested 15 h later with an OD$_{600}$ of 5.5.

For the uniform [$^{2}$H, $^{13}$C, $^{15}$N] labeling, a frozen stock was prepared beforehand by highly expressing cells of BL21(DE3)/pET28a-AS using the "double colony selection" method of Sivashanmugam et al.[72] This stock was used to streak an LB plate containing 70% D$_2$O and 40 µg/ml kanamycin which was grown overnight. The colonies were then used to inoculate LB-kanamycin in 70% D$_2$O which was grown at 37 °C to an OD$_{600}$ of 3.6. The cells were gently pelleted and resuspended in an equal volume of 99% D$_2$O SHD medium (modified from the Sivashanmugam et al. optimized High cell-Density medium[72]), containing 50 mM Na$_2$HPO$_4$, 25 mM KH$_2$PO$_4$, 10 mM NaCl, 5 mM MgSO$_4$, 0.2 mM CaCl$_2$, 8 g/L [$^{2}$H, $^{13}$C]glucose, 1 g/L [$^{15}$N]ammonium chloride, 10 mL/L [$^{2}$H, $^{13}$C, $^{15}$N]Bioexpress, 2.5 mL/L BME vitamins, 0.25 mL/L Studier trace metals, 90 µg/mL kanamycin, and was adjusted to pD 8.0 with NaOD[73]. Growth was continued with 40 ml culture per baffled 250 ml flask at 25 °C and 250 rpm shaking, to promote aeration. At an OD$_{600}$ of 4.3, expression was induced with 0.5 mM IPTG. The culture was harvested after 15 h with a final OD$_{600}$ of 10.6.

Protein purification was done in E. Coli BL21(DE3)/pET28a-AS[74]. Briefly, cells were lysed chemically in the presence of Turbonuclease (Sigma) to digest nucleic acids. Purification began with a heat denaturation of the cleared lysate, followed by ammonium sulfate precipitation[75]. The resolubilized protein was bound to QFF anion exchange resin (GE Healthcare Life Sciences, Marlborough, MA) and eluted using a linear gradient of 0.2–0.6 M NaCl. Fractions containing Asyn monomer, which eluted at about 0.3 M NaCl, were pooled, concentrated, and run over a 26/60 Sephacryl S-200 HR gel filtration column (GE Healthcare Life Sciences) equilibrated in 50 mM Tris-HCl, 100 mM NaCl, pH 8 buffer. Fractions were pooled, concentrated to ~20 mg/ml a-synuclein, and dialyzed at 4 °C into 10 mM Tris-HCl pH 7.6, 50 mM NaCl, 1 mM DTT, and stored at a concentration of ~14 mg/mL at −80 °C until use. Yields were 95 mg purified AS protein/L growth medium for the uniform [$^{13}$C, $^{15}$N] labeled monomer, 288 mg/L for the uniform [$^{2}$H, $^{13}$C, $^{15}$N] labeled monomer, and 217 mg/L for the [2-$^{13}$C] glycerol, $^{15}$N labeled monomer.

## Amplification of isotopically labeled LBD-amplified fibrils for SSNMR studies

SSNMR experiments relied on large quantities of amplified fibrils. We selected postmortem case LBD1 for SSNMR characterization. Multiple replication tubes of LBD1 amplification were setup using uCN and uCDN Asyn monomers. The protocol to generate LBD amplified fibrils was similar to that described above, except that 50k Amicon ultra filtered isotopically labeled Asyn monomer was used. In addition, the amplification reactions were supplemented by control fraction preparation derived from E.Coli transformed with an empty expression vector. This control fraction was purified with the same protocol as the natural abundance Asyn monomer. Amplified fibrils were harvested by ultracentrifugation, washed with either H$_2$O or D$_2$O, dried under a gentle stream of nitrogen gas, packed into 1.6 mm rotors (Revolution NMR, LLC) and hydrated to approximately 40% with either H$_2$O or D$_2$O, as described[76].

## Amplification of MSA fibrils from MSA insoluble fraction seeds

We produced MSA amplified fibrils from the caudate brain region, matching the brain selected for LBD amplification. The protocol to generate MSA amplified fibrils was similar to the LBD amplified fibrils until the first 4 cycles. Further expansion was performed by bringing up 60 μl of 4th cycle MSA amplified fibrils to 100 μl and sonicating the mixture for 1 min at amplitude 50. To this mixture, Asyn monomer was added to a final concentration of 2 mg/ml in a final volume of 400 μl in fibril buffer. This mixture was quiescently incubated at 37 °C for 2 days (5th cycle). To expand further, the 400 μl of MSA amplified fibrils after 5 cycles were sonicated for 1 min at amplified 50 and additional 2.1 mg/ml of Asyn monomer was added to bring the total volume to 800 μl. This mix was incubated at 37 °C for 2 days to complete the 6th cycle of incubation. The increased monomer concentration (2.1 mg/ml instead of 2 mg/ml) was calculated based on the loss of free Asyn monomer due to its incorporation into amplified fibrils. After 6 cycles, fibrils were stored at 4 °C until use. In parallel, insoluble fraction from control tissue samples were also amplified under the similar conditions used for LBD amplified fibrils.

## Preparation of Asyn fibrils in Tris buffer (IV$_{Tris}$ fibrils)

Fibrils were assembled in vitro in a 2 ml tube (Catalog 430659, Corning,) by incubation of 2 mg/ml recombinant Asyn monomer in 20 mm Tris-HCl, pH 8.0, 100 mm NaCl buffer in a total volume of 1500 μL. The incubation was carried out at 37 °C with continuous shaking at 1000 rpm in a Thermomixer (Eppendorf) for 72h[39,42]. After IV$_{Tris}$ fibrils were formed, tube was moved to 4 °C until use.

## Radioligand quantification of amplified fibrils

Radioligand binding assay was used to estimate the specificity and quantify the growth of LBD amplified fibrils at the end of the amplification process. A modified version of homologous competition binding assay with the radioligand [3H]-BF2846 (Specific Activity: 73.3 Ci/mmol) was utilized to assess the fibrils after 4 and 6 cycles of amplification. We assumed equal affinity and binding site density of the radiotracer to LBD and control amplified fibrils. A fixed volume of amplified fibrils (LBD and control) were added to either 300 nM cold BF2846 (made in DMSO) or with no cold (DMSO only) along with 2 nM of [3H]-BF2846. The total volume of reaction mixture was 150 μl in 30 mM Tris-HCl, pH 7.4 plus 0.1% BSA buffer. The mix was incubated for 2 h at 37 °C with shaking (1000 rpm) in an Incu-Mixer MP2 (Catalog H6002, Benchmark Scientific). Samples were transferred to Multiscreen FB Filter plates (Catalog MSFBN6B50, MilliporeSigma) and washed three times with cold (4 °C) 30 mM Tris-HCl, pH 7.4 plus 0.1% BSA buffer. The glass fiber filters containing fibril bound radioligand were removed and counted overnight in a PerkinElmer (1450-021) Trilux MicroBeta Liquid Scintillation counter using 150 μl of Optiphase Supermix cocktail (Perkin Elmer). All data points were done in triplicate. The displacement counts between the two different cold concentrations was converted to concentration of fibrils and used for comparison of specificity between disease and control amplifications. To determine concentration of LBD amplified fibrils, specific [3H]-BF2846 binding was determined from $n = 3$ wells with and without 0.3 μM cold BF2846. Specific binding was then converted to μmol of bound ligand using the specific activity. Fractional occupancy was calculated based on previously measured K$_d$, and then previously measured B$_{max}$ was used to calculate the concentration of Asyn monomeric units present in fibrils. This radioligand concentration estimation method allowed rapid assessment of amplifications and further validation of this method was performed using Micro BCA and Guanidine hydrochloride (GdnHCl)-A$_{280}$ assay.

## Concentration estimation of LBD and control amplified fibrils by Micro BCA assay

We utilized the micro-BCA assay to estimate the amount of fibrils. After 6 cycles, amplified fibrils were centrifuged at 21,000 xg for 15 min at

4 °C to separate fibrils from monomer. The concentration of Asyn monomer in the supernatant was determined by the Micro BCA protein assay (Thermo Scientific Pierce Micro BCA kit, catalog no. 23235) according to the manufacturer's instructions, using the manufacturer-supplied bovine serum albumin (BSA) for the standard curve. At the same time, total sample containing fibrils plus unincorporated monomer was also assessed. The measured decrease in Asyn monomer concentration between total and supernatant was used to determine the concentration of fibrils.

## Concentration estimation of LBD amplified fibrils by Guanidine hydrochloride (GdnHCl)-A$_{280}$ assay

A 30 μl volume of LBD amplified fibrils after 6 cycles of amplification was centrifuged at 100,000xg for 15 min at 4 °C. The pellet was further washed with 150 μl of 20 mM Tris-HCl, pH 8.0 plus 100 mM NaCl and centrifuged at 100,000×g for 15 min at 4 °C to reduce unincorporated Asyn monomer. Pellets were resuspended in 30 μl of 6 M GdnHCl solution in 50 mM Tris-HCl, pH 8.0 and sonicated for 1 min at amplitude 20 in a bath sonicator. The fibrils were then incubated with 6 M GdnHCl for 2 h at room temperature with shaking (750 rpm) to denature the fibrils. A NanoDrop Microvolume Spectrophotometer (ThermoFisher) was used for A280 measurements using an extinction coefficient of 5960 M$^{-1}$ cm$^{-1}$ and molecular weight of 14.4 kDa for Asyn.

## Characterization of amplified fibrils via negative stain TEM

Negative staining of the different fibril conformers was performed by applying a given fibril preparation to ultrathin Carbon 300 mesh Gold grids (Catalog 01824 G, Ted Pella). The grids were negatively glow discharged (13 mA, 45 s) using GloQube glow discharge system (Model #025235 EMS). A 10 μl fibril sample at appropriate dilution was applied to the glow discharged grid for 5 minutes with carbon side facing the sample drop. Post-sample incubation, the grid was washed (6 times) with 50 μl of H$_2$O, washed once with 50 μl of 0.75% uranyl formate and stained with 50 μl of 0.75% uranyl formate for 3 min. The grids were blotted using filter paper, leaving a small amount of stain to air dry on the grid surface. Grids were imaged on a JEOL 1400 TEM operating at 120 kV to visualize the negatively stained fibrils.

To estimate the diameter of amplified fibrils, 30 μl samples of LBD amplified fibrils after 2, 4, and 6 cycles of amplification were sonicated for 1 min at amplitude 20 in a bath sonicator. These preparations were applied to ultra-thin grids and negatively stained as described above. Several micrographs at 60,000× magnification were collected and the "straight line" function in ImageJ was used to measure the diameter along the length of the fibrils. In parallel, negative staining of IV$_{Tris}$, MSA (after 6 cycles) and control (after 6 cycles) amplified fibril samples was also performed.

## SDS-PAGE gel electrophoresis of LBD, MSA, and control amplified fibrils

LBD, MSA, and Control amplified fibrils at 2, 4, and 6 cycles were centrifuged at 100,000×g for 15 min at 4 °C to remove unincorporated Asyn monomer. IV$_{Tris}$ fibrils were treated similarly. Pelleted fibrils were resuspended in DPBS plus 0.1% Triton X-100 and sonicated for 2 min at amplitude 20 to disperse the fibrils. Samples were then resuspended in 1× XT sample buffer containing 5% β-mercaptoethanol and boiled for 5 min at 95 °C. Samples were run on 12% Criterion™ XT Bis-Tris Protein Gel (Catalog 3450118, Bio-Rad) with XT MES running buffer. Precision Plus Protein Unstained Standards (Catalog 1610363, Bio-Rad) were used as a protein standard. The gel was stained overnight using SYPRO™ Ruby Protein Gel Stain (Catalog S12001, Invitrogen) according to manufacturer instructions and imaged on the Syngene Gel Doc imaging system. Due to the presence of low quantities of fibrils after 6 cycles in control cases, amplified fibrils from three control cases were pooled and concentrated 7 times compared to the LBD amplified fibril sample to get similar amounts for loading onto the SDS-PAGE gel. For

similar reasons, 2nd and 4th cycle control amplified fibrils were concentrated 3 and 6 times compared to the 2nd and 4th cycle LBD amplified fibril samples for SDS-PAGE.

## Proteinase-K digestion of LBD, MSA, and control amplified fibrils after 6 cycles of amplification

LBD, MSA, and control amplified fibrils after 6 cycles of amplification were spun down at 100,000×g for 15 min at 4 °C to remove unincorporated Asyn monomer. Fibrils were then resuspended in DPBS plus 0.1% Triton X-100 and sonicated for 2 min at amplitude 20 to disperse the fibrils. A total of 24 μg of LBD, MSA, and pooled control amplified fibrils were digested with 1.2 μg of Proteinase-K (Catalog P2308, Sigma-Aldrich) for 30 min at 37 °C in PCR tubes. Digestion of $IV_{Tris}$ and 50k purified Asyn monomer was done in parallel. The digestion was stopped by adding an equal volume of 2× XT sample buffer containing 10% β-mercaptoethanol. The samples were boiled and run on the 12% Bis-Tris gel and SYPRO Ruby stained as described before. Due to the presence of low quantities of fibrils after 6 cycles in control cases, amplified fibrils from three control cases were pooled and concentrated 12 times compared to the LBD amplified fibril sample for Proteinase-K digestion.

## Western blot of LBD amplified fibrils

LBD amplified fibrils (4th cycle) were prepared and run on 12% Criterion™ XT Bis-Tris Protein Gel as described before for SDS-PAGE. Full-length Asyn monomer was also run alongside the LBD amplified fibril samples as control. The samples were then transferred to PVDF membrane and blocked with 5% milk in PBS-0.2% Tween20 for 1 h and incubated with the primary antibody in blocking solution overnight at 4 °C. We utilized Syn303 (Catalog 824301, Biolegend, epitope: Asyn N-terminus, amino acids (aa) 1-5), Syn1(Catalog 610787, BD Transduction Laboratories Clone 42, epitope: Asyn aa91-99), Syn211 (Catalog 32-8100, Invitrogen, Epitope: Asyn aa121-125) and 13G5 (in-house, epitope: Asyn aa120-131) at 1 μg/ml final concentration to detect Asyn. After primary antibody incubation, membranes were washed three times with PBS-0.2% Tween20 and incubated with secondary goat anti-mouse HRP antibody (Catalog 7076, Cell Signalling Technology) at 1:5000 dilution for 1 h. Membranes were washed again three times and probed with ECL reagent (Cytiva RPN2209, Millipore Sigma) and auto-exposed using the Syngene Gel Doc imaging system.

## Cryo-EM grid preparation and imaging

Amplified Asyn fibril samples were prepared for cryo-EM imaging on Quantifoil holey carbon grids (R2/2 300 mesh copper) by plunge freezing using a Vitrobot Mark IV (ThermoFisher Scientific, Brno, CZ). Grids were plasma cleaned for 1 min using H2/O2 plasma in a Gatan Solarus 950 (Gatan, Pleasanton, CA) prior to plunge freezing. The sample chamber of the Vitrobot was set to 4 °C and 95% humidity. 3 μL of sample was applied to the grid surface. After 20 s of incubation, the grids were blotted for 2 s with a blot force of 4 and plunge frozen into liquid ethane. Single particle cryo-EM imaging was performed using a Cs-corrected ThermoFisher Titan Krios G3 cryo-electron microscope (ThermoFisher Scientific, Eindhoven, NL) operating at a 300 kV accelerating voltage. EPU (ThermoFisher Scientific, Brno, CZ) was used for image acquisition using a Falcon IV direct electron detector (ThermoFisher Scientific, Eindhoven, NL) at a magnification of 75,000x which corresponds to a 0.9 Å pixel size. Movies were acquired for 8.38 s and used 48 total frames. The total dose was 53.04 electrons per Å2 and the total dose per frame was 1.105 electrons per Å2 per frame. The defocus range was −1 to −2.4 μm.

## Cryo-EM image processing

Cryo-EM data was processed using helical reconstruction methods in RELION[51]. Movies were motion-corrected and dose-weighted using MOTIONCOR2[77]. Motion-corrected micrographs were then to estimate

the contrast transfer function (CTF) using GCTF[78]. Asyn filaments were first picked manually to generate 2D templates via 2D Classification in cryoSPARC. Once low-resolution templates were obtained, they were used to train the Filament Tracer helical auto-picking feature. The final selection of particles picked by Filament Tracer were extracted using a box size of 280 pixels with a 0.9 Å pixel size. Initial particles were reduced to a final set of ~63k after several rounds of 2D classification in cryoSPARC. This particle set was then exported into RELION, via csparc2star.py, for an additional round of 2D classification.

## Cell culture experiments

To assess the fidelity of amplification, we utilized HEK293T "biosensor" cell line stably expressing Asyn (A53T)-CFP/YFP fusion proteins[47]. Glass coverslips were coated with poly-d-lysine hydrobromide (Sigma) overnight at room temperature, then washed 3 times with sterile water and allowed to air-dry for 2 h. Biosensor cells were plated in a 24-well plate at 75,000 cells per well and grown for 1 day in 400 μl of cell culture media. Biosensor cells were then treated with amplified fibrils, insoluble fraction, or the soluble fractions derived from the amplified fibrils and the postmortem tissue. Seeding samples were diluted in Opti-MEM (Gibco) to a final volume of 50 μl, then, sonicated for 1 min at amplitude 50 in the bath sonicator. To this, 3 μl of Lipofectamine 3000 in 50 μl Opti-MEM was added and the mixture incubated for 30 min at room temperature for complex formation. This 100 μl of fibrils plus Lipofectamine mixture was added dropwise to 400 μl of the culture medium in each well of a 24-well plate. For insoluble and soluble fraction derived from tissue, we added 5 μl per well. Pilot experiments were utilized to determine volume amounts of LBD and MSA fibril seed samples required to produce aggregates in 10-30% of the cells, before proceeding to experiments for quantitative analysis. This approach was utilized to obtain reliable quantification of seeding in the setting of significantly different seeding efficiencies of LBD and MSA fibrils in pilot experiments. Concentrations of the fibril samples were separately measured after the biosensor seeding experiments. For LBD amplified fibril samples, concentrations between 9–26 nM were used. For MSA amplified fibril samples, concentrations between 0.1–0.15 nM were used. After 72 h, biosensor cells were fixed with 4% paraformaldehyde (EMS) for 15 min and washed with Dulbecco's PBS three times. Coverslips were then mounted on glass slides using Fluoromount-G containing DAPI (Southern Biotechnology). Coverslips were imaged on a Nikon Eclipse TE2000-U fluorescence microscope using a Nikon Plan Fluor ×40/0.75 and a 10× objective. Images were collected using Metamorph 6.1 (Molecular Devices) software package and assembled in Photoshop CS3 (Adobe). Natural abundance Asyn monomer amplified samples (LBD3, LBD4, LBD5, MSA1, MSA2, and MSA4) were selected for the cell culture experiments. To quantify the number of inclusions formed after seeding of LBD and MSA amplified fibrils, fluorescent inclusions were quantified in images from 40× objective using imageJ. The number of fluorescent aggregates per well were calculated based on area and normalized to the amount of LBD and MSA amplified fibrils added to get the number of inclusions per pg of fibrils.

## Solid-state NMR spectroscopy

Magic-angle spinning (MAS) SSNMR experiments were performed at magnetic field of 11.7 T (500 MHz ¹H frequency) or 17.6 T (750 MHz ¹H frequency) using Varian NMR (Walnut Creek, CA) VNMRS spectrometers. Spinning was controlled with a Varian MAS controller to $11,111 \pm 30$ Hz or $22,222 \pm 15$ Hz (500 MHz ¹H frequency) and $16,667 \pm 15$ Hz or $33,333 \pm 30$ Hz (17.6 T), with two exceptions indicated in Supplementary Table 6. Sample temperature during SSNMR data collection was $10 \pm 5$ °C. The 11.7 T magnet was equipped with a 1.6 mm HCDN T3 probe (Varian), with pulse widths of about 1.8 μs for ¹H and ¹³C, and 3.2 μs for ¹⁵N. The 17.6 T magnet was equipped with a HXYZ T3 probe (Varian) tuned to HCN triple resonance mode with pulse widths of 1.9 μs for ¹H, 2.6 μs for ¹³C, and 3.0 μs for ¹⁵N. All experiments utilized

[1]H-[13]C or [1]H-[15]N tangent ramped CP[79] and -100 kHz SPINAL-64 decoupling during evolution and acquisition periods[80,81]. Where applicable, SPECIFIC CP was used for [15]N-[13]Cα and [15]N-[13]C' transfers[82]; [13]C-[13]C homonuclear mixing was performed using DARR[83], RFDR[84,85], or PAR[86]; and water suppression was done using MISSISSIPPI[87]. FS-REDOR was performed according to Jaroniec et al.[88] The reference spectrum was acquired with a 630 µs Gaussian pi pulse centered on resonance with the Glu CDs, while the REDOR dephasing was performed using 9 µs square pi pulses placed every half rotor period on the [15]N channel, except for a 540 µs Gaussian pi pulse in the middle of the dephasing period placed on resonance with the Lys NZs. Chemical shifts were externally referenced to the downfield peak of adamantane at 40.48 ppm[89]. NUS schedules using biased exponential sampling were prepared using the nus-tool application in NMRbox[90]. Data conversion and processing was done with NMRPipe[91]. NUS data was first expanded with the nusExpand.tcl script in NMRPipe, converted, and processed using the built-in SMILE reconstruction function[92]. Peak picking and chemical shift assignments were performed using SPARKY 3.

## Mass-per unit length measurements with electron microscopy
TEM grids were prepared using a 1 min oxygen plasma cleaning treatment (Harrick Plasma Cleaner PDC-32G, at low power). One 10 µl droplet of Asyn fibril suspension and three 10 µl droplets of ultrapure water for each grid were pipetted onto a clean sheet of Parafilm. Freshly plasma-cleaned TEM grids were inverted onto a sample droplet and rested for 60 s. Excess sample solution was blotted away with filter paper, and grids were placed onto each of the three droplets of water and blotted again in quick succession, to rinse away excess salt. Grids were then rested on a droplet of tobacco mosaic virus (TMV) suspension, prepared by diluting a stock solution to 0.12 mg/mL. TMV was used to calibrate electron density in each image[93–95]. Imaging was done on a JEOL 2100 Cryo-TEM using an electron accelerating voltage of 80 kV. Micrographs were collected in the tilt-beam geometry using the third objective aperture, an exposure time of 3–5 s. Short, non-overlapping segments of TMV and Asyn fibrils in each image were then selected using the helixboxer function of EMAN2[96] and exported to the MpUL-multi program for quantification of MPL statistics[97]. Segments were only chosen from TMV and Asyn fibrils when they were distinguishable in both tilt-beam and accompanying bright-field micrographs.

## Xplor-NIH structure calculations
Automated structure calculation and refinement was done using the PASD protocol[57,58] in Xplor-NIH[59]. Four SSNMR spectra were used for assignment of long-range correlations: (1) 2D [13]C-[13]C with 12 ms PAR mixing collected on the uCN sample; (2) 3D [13]C-[13]C-[13]C with 1.9 ms RFDR then 12 ms PAR mixing collected on the uCN sample; (3) 3D [15]N-[13]Ca-[13]CX with 12 ms PAR mixing collected on the uCN sample; and (4) 2D [13]C-[13]C with DARR mixing collected on the diluted sample. Peaks were picked manually for 2Ds or using restricted peak picking for 3Ds in SPARKY 3 then filtered against the resonance list to remove any peaks without at least one assignment possibility using an in-house Python script. These lists were used directly as input for the automated generation of peak assignments using PASD.

Calculations were performed using the strict symmetry facility in Xplor-NIH[59]. Based on the model of two protofilaments that are symmetrically arranged, only a single protomer subunit was explicitly simulated, while four additional subunits were generated by translation along the z-axis by the centroid position of the first protomer, and five were generated by rotation of 180 degrees about the z-axis followed by translation. This approach simplified restraint generation because the only interactions that were critical to simulate were those between the first protomer (A1) and one generated by translational symmetry (B1), and those between the first protomer (A1) and one in-plane generated by rotational symmetry (A2). A restraint with an atom-

atom ambiguity of one in terms of the number of possible resonance assignments therefore had a total ambiguity of five (intramolecular, A1-A1; intermolecular, A1-B1, B1-A1, A1-A2, A2-A1). The intramolecular B1-B1 and A2-A2 were identical by symmetry and therefore were not included as assignment options. Likewise, the intermolecular/interfilament B1-A2 and A2-B1 were also excluded.

To match cross-peak positions to assigned chemical shift resonances, the following tolerances were used: 0.25–0.35 ppm for indirect [13]C dimensions, 0.35 ppm for indirect [15]N, and 0.20–0.25 ppm for direct dimensions. Automated chemical shift correction was not performed. Distances for all three PAR mixing data sets were binned as strong (up to 4.5 Å), medium (up to 6.0 Å), weak (up to 7.0 Å), and very weak (up to 8.5 Å) and the DARR were binned as strong (upto 4.5 Å), medium (up to 5.5 Å), weak (up to 6.5 Å), and very weak (up to 7.5 Å). Long-range assignment options for the isotopically diluted sample data were assumed to be exclusively intramolecular, whereas all possibilities were included as ambiguous for the uCN and uCDN sample data. After the initial matching of peak lists to the resonance list, the network filter was applied. In addition to assigning likelihoods of 0 or 1 to peak assignments based on the network filter analysis, peaks with intraresidue assignment possibilities or peaks with at least one peak assignment that was short-range (differed in primary sequence by less than 3 residues), or peaks with more than two possible atom-atom peak assignments were removed. This reduced file sizes and computation time by significantly reducing the number of interactions that need to be computed.

The initial coordinates for each structure calculation were those of a geometry-optimized protomer with backbone dihedral angles satisfying the TALOS-N restraint list, and truncated to residues 30–103, excluding the disordered N- and C-terminal regions that are not in the fibril core. Calculation of 500 structures for each pass (pass 2, pass 3, and pass 4) was performed according to the protocol in Supplementary Table 11. In addition to the PASD peak lists, a few manually assigned distance restraints were included in all structure calculations to help guide simulated annealing. These included backbone hydrogen bond registry restraints based on non-exchanging [1]H signals observed in the uCDN sample, seven restraints that were hypothesized to be exclusively intrafilament based on some preliminary structure calculations, and three restraints that were hypothesized to be exclusively interfilament using similar logic. Statistics were collected on the 200 lowest energy structures for each pass. Peak assignment likelihood reanalysis following each PASD structure calculation pass was done using the 20 best-converged structures from the low energy fraction. Pass 2 and pass 3 were run using only peaks with an atom-atom peak assignment ambiguity of 1 or 2. Likelihood reanalysis based on the pass 3 structures was done using the pass 1 peak lists, which re-introduced the originally higher ambiguity peaks, now at significantly lower ambiguity. Following pass 4, the high-likelihood restraints (≥0.9) were converted into Xplor-NIH format and used in a pass 5 standard simulated annealing calculation employing 192 structures. The 19 structures with the lowest energy were used for an additional likelihood reanalysis and as input into a structure-dependent PASD refinement (pass 4), which was otherwise identical to pass 3 and pass 4. A final single-assignment restraint list was generated using a few iterative rounds of regular simulated annealing calculations starting with the Xplor-NIH restraint lists based on pass 5, manual adjustment to the restraint lists based on study of the violation statistics, and automated peak assignment likelihood reanalysis. At this stage, medium range peak assignments (including restraints between neighboring adjacent in sequence from the initial matching) were also incorporated to help with optimization of local side chain orientations. The final refinement calculation also included an intrafilament salt bridge modeling restraint between residues E35 and K80 based on the detection from FS-REDOR (Supplementary Fig. 18) that the structure contains at least one salt-bridge between a Glu CD and a Lys NZ and the fact that this salt

bridge was observed in the majority of converged structures even without this modeling restraint being present.

## Assessing structural agreement from PASD assignments

NMR peak lists were used to assess agreement of the structures with atomistic assignments. Peak lists with associated NMR assignments were loaded into Xplor-NIH as PASD potentials and assignments that did not agree with inter-atomic distances in a provided structure to within 0.7 angstroms were removed. All assignments that agreed with an inter-atomic contact to within 0.7 angstroms were extracted and compiled on a per-residue basis. Satisfied assignments are shown in Fig. 9d, e as heat maps.

## Molecular dynamics (MD) simulations and analysis

An all-atom MD simulation of a two-fold symmetric Asyn fibril (2 × 20-mer) was performed using NAMD 3.0[98]. The CHARMM36m protein force field was applied to the fibrils[99]. We simulated the fibril in an explicit water box of size 180 x 100 x 180 Å3 with an ion concentration of 0.15 M (NaCl solution) using an NPT ensemble with $N = 306450$, $P = 1.01325$ bar and T = 310 K. The system was subjected to Langevin dynamics with damping constant $γ = 1.0$ ps-1 and the Nosé-Hoover Langevin piston method[100,101] was employed to maintain constant pressure. The MD integration time step was set to 2 fs. To evaluate long range electrostatic interactions, particle mesh Ewald[102] was used in the presence of periodic boundary conditions. Nonbonded van der Waals (vdW) interactions were calculated with the Lennard-Jones (12, 6) potential with a cutoff distance of 12 Å. A smoothing function was introduced at 10 Å to gradually truncate the vdW potential energy at the cut-off distance. 14 Å cutoff distance was used to identify the atom pairs for vdW interaction. The system was first energetically minimized for 1000 steps by conjugate gradient method followed by a 10 ns equilibration of water and ions in the presence of positional restraints (force constant: 1 kcal/mol Å2) on Cα atoms of the fibril. The system was then simulated for 200 ns without any restraints. Simulation trajectories were collected every 50 ps.

## Mutagenesis for C2-mutant-Asyn

The plasmids C2-A53T-Asyn, C2-E46K-Asyn and C2-H50Q-Asyn were generated in the previous study[39]. For generating plasmids of C2-S87K-Asyn, C2-87Q-Asyn, C2-G68Q-Asyn and C2-A76T-Asyn, site directed mutagenesis was performed using the Q5 Site-Directed Mutagenesis Kit protocol (New England Biolabs) on the pRK172-C2-WT-Asyn plasmid using the following primers: S87K mutation, 5′-GGGAGCAGGGaaaATTGCAGCAG-3′ (forward primer) and 5′-TCCACTGTCTTCTGGGCT-3′ (reverse primer); S87Q mutation, 5′-GGGAGCAGGGcagATTGCAGCAGCC-3′ (forward primer) and 5′-TCCACTGTCTTCTGGGCT-3′ (reverse primer); G68Q mutation, 5′- AAATGTTGGAcagGCAGTGGTGACGG-3′ (forward primer) and 5′- GTCACTTGCTCTTTGGTC-3′ (reverse primer) and A76T mutation, 5′-GGGTGTGACAaccGTAGCCCAGAA-3′ (forward primer) and 5′- GTCACCACTGCTCCTCCA-3′ (reverse primer). All mutations were confirmed by sequencing of the entire Asyn coding region. C2-mutant-Asyn was produced using the same method as the natural abundance WT Asyn monomer.

## Synuclein polymorph analysis by relative kinetics (SPARK) assays

We developed a sensitive assay that uses the fluorescein arsenical dye FlAsH-EDT$_2$ (Catalog T34561, Invitrogen TC-FlAsH™ in-cell tetra-cysteine tag detection kit) to measure fibril growth rates in the presence of C2-WT and mutant C2-Asyn monomer. The SPARK assay was similar to FlAsH seed growth experiments demonstrated in our previous publication[39]. Amplified fibril samples at 1.5 μM concentration in fibril buffer plus 0.1% Triton X-100 were sonicated for 5 min at amplitude 50 in the cup horn sonicator (Qsonica) and then mixed with 0.5 mg/ml of C2-Asyn monomer (C2-WT-Asyn and mutant-C2-Asyn) in

a total volume of 25 μl. The seeds plus monomer mixture were quiescently incubated for 3 h at 37 °C in Corning Black 96-well plates (Catalog 07-200-762, Fisher). After 3 h, FlAsH assay mixture consisting of 3.5 mm tris(2-carboxyethyl)phosphine, 1 mm EDT, 1 mm EDTA, 25 nm FlAsH-EDT$_2$, and 200 mM Tris-HCl, pH 8.0 was added and the mixture incubated for additional 1 h at room temperature. FlAsH fluorescence was detected in a BioTek plate reader using a 485/20-nm excitation filter, a 528/20-nm emission filter, top 510-nm optical setting, and gain setting 100. Data was normalized to fibril growth rates with C2-WT-Asyn monomer. We selected LBD and MSA amplified fibrils after 6 cycles (case LBD1, LBD2, LBD3, LBD4, MSA1, MSA2, MSA3, and MSA4) for SPARK assays. To analyze SPARK data from LBD amplified fibrils, we utilized unpaired, two-tailed t-tests (Welch) and corrected for multiple comparisons with the Holm-Bonferroni method using a significance level of 0.05.

## Reporting summary

Further information on research design is available in the Nature Portfolio Reporting Summary linked to this article.

## Data availability

The datasets generated during and/or analysed during the current study are available from the corresponding author on request. The ten lowest energy structural models have been deposited in the Protein Data Bank (PDB) under accession number 8FPT. NMR data have been deposited in the Biological Magnetic Resonance Bank: BMRB 31068 (Structure of Alpha-Synuclein fibrils derived from Human Lewy Body Dementia Tissue). Molecular dynamics simulation scripts and data have been submitted in the DRYAD public repository, [https://doi.org/10.5061/dryad.tx95x6b4z]. Backbone assignments are provided in BMRB ID 51678. Chemical shifts assignments have been submitted to the BMRbig, under the entry ID: BMRbig83. Additional source data is included in source data file. Source data are provided with this paper.

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

## Acknowledgements

Support for this work was provided by: grants from the Michael J. Fox Foundation; NIH grants NS110436, NS097799, and NS075321 from the National Institute of Neurological Disorders and Stroke and National Institute on Aging; P41GM136463 from the National Institute of General Medical Sciences; the American Parkinson Disease Association (APDA) Advanced Research Center for Parkinson Disease at Washington University in St. Louis; the Greater St Louis Chapter of the APDA. We are grateful for the technical support of the Betty Martz Laboratory for Neurodegenerative Research at Washington University in St. Louis (PK). We acknowledge the assistance of Dr. Rocio Bengoechea with the immunoblots. CS was supported by the Intramural Research Programs of NIDDK and NHLBI at the National Institutes of Health. OW was supported by the Molecular Biophysics Predoctoral Training Grant T32GM130550 from NIGMS. CB is supported by the NIH Ruth L. Kirschstein Fellowship (F32-GM149118) from NIGMS. Figure 1a created with Biorender.com.

## Author contributions

Study design and organization: D.D., A.B., C.R., P.K. Recruitment and collection of autopsy cases: R.P., J.P., P.K. Acquisition of data: D.D., A.B., C.B., D.B., J.L., J.O., M.R., S.S., J.S. Analysis and interpretation of data: D.D., A.B., C.B., K.B., I.G., M.M., M.R., Z.S., S.S., B.S., J.S., O.W., Q.C., J.F., C.S., E.T., C.R., P.K. Drafting of the manuscript: D.D., A.B., C.B., K.B., M.R., C.R., P.K. Critical revision of the manuscript for important intellectual content: All authors.

## Competing interests

Authors specified below have a patent application pending titled "Tissue-Seeded Fibrils and Methods of Making and Using Same". Patent applicant: Washington University in St. Louis, Name of inventors: Paul T. Kotzbauer, Dhruva D. Dhavale, Rebecca Miller and Jennifer Y. O'Shea, Application number: 17/858817, Status of application: Pending. The patent application covers the process of generating amplified fibrils, its methods and composition. Other authors declares no competing interests.

## Additional information

**Supplementary information** The online version contains Supplementary Material available at https://doi.org/10.1038/s41467-024-46832-5.

