## [Peer Review File · Nature Communications]

Reviewers' Comments:

Reviewer #1:

Remarks to the Author:

Previous studies using cryo-EM have shown that alpha-synuclein filaments from PD and DLB brains comprise a single protofilament. The limited helical twist of these filaments however has made the structure of these filaments difficult to resolve.

This study advances the field by 1) developing a method to amplify 5 mg of fibrils from DLB brain extracts, 2) using Tx-100 as a more gentle extraction which preserves the alpha-synuclein fibrillar structure, and 3) using solid state NMR with higher resolution (relative to cryo-EM) to demonstrate that the majority of DLB fibrils comprise two protofilaments with very little twist.

Interestingly, the authors use the structure to identify residues that reduce fibril growth for all three fibril types and selectively for LBD fibrils.

The biggest concern is that the structure appears to be derived from only one LBD case. Many other studies compare across multiple cases. For example, the Yang et al paper compared structures from cortex from 3 DLB, 1 PD, 2 PDD. There also appear to be no statistics. How can the authors be sure these findings are consistent across multiple cases and this was just an artifact from one brain with a relatively long (14 hr) postmortem interval? Also, what were the brain regions used for the control, LBD and MSA cases? It is likely that different conformations exist across different areas of the brain.

It is also possible that one form of filament amplifies more than others because it is more concentrated and the amplification reaction favors this more stable, dominant form. It is also possible that important assemblies are lost in Tx100 fraction and that focusing on the insoluble fraction selects for only one type of LBD filament. These are important caveats for this and previous studies using amplification methods.

The biosensor cells with high levels of expressed A53T- α -synuclein are okay. However, there is no quantitation of the percentage of cells with aggregates. Also, it would have a greater impact on the field to see seeding in primary neurons which have high concentrations of normal synuclein at synapses and do not need a lipid carrier for entry of the fibrils into the neurons. Alpha-Synuclein expressed in immortalized cells is not converted to an aggregated form as easily as alpha-synuclein in neurons. The "soluble" DLB filaments could seed alpha-synuclein aggregates in primary neurons

Also, while not necessary for this study, it is important to note that soluble alpha-synuclein assemblies could show different morphology compared to the insoluble seeds. Such data would also be useful to the field for determining the impact of DLB assemblies on neuronal function and spread in the nervous system.

Furthermore, according to the methods "For LBD amplified fibrils samples, concentrations between 26 nM to 1 nM were used. For MSA amplified fibril samples, concentrations between 1.5-0.1 nM was used." Ideally a concentration curve should be performed with quantitation of aggregates quantified at the different concentrations. The BCA assay is an indirect and not correct method of measuring the synuclein concentration. The fibrils can be dissociated into monomer with guanidine hydrochloride and the protein then either measure using BCA or ideally, using Beer-Lambert law.

Also, unlike for the insoluble form of synuclein, for soluble tissue fraction, Tx-100 was not used and thus some assemblies likely were not extracted and lost to the pellet. It is convincing (although quantitation would help solidify these data) that the "soluble" fraction from MSA seeds aggregates in the biosensor cells, and the LBD "soluble" fraction does not. It is possible some assemblies would be available in the Tx-100 extracted "soluble" fraction—this fraction could be dialyzed to remove triton. While not necessary for this paper, it would be interesting to know if some "Tx-100" soluble synuclein assemblies have seeding activity.

The authors state in the introduction that the Lewy fold found by cryo-EM is distinct from MSA or recombinant fibrils, but this is misleading. Amino acids 61-72 are the same in both MSA and recombinant filaments. Also, amino acids 32-41 and 70-82 are the same in the Lewy fold and recombinant fibrils.

For figure 1d, did the authors quantify the percentage of fibrils with paired helical filaments and straight fibrils across all preparations from multiple fields? It is difficult to make conclusions from EM snapshots.

Were the cases used pathologically confirmed to show Lewy pathology, neurofibrillary tangles, amyloid beta? Were any of the cases genotyped for ApoE4?

Can the authors discuss the significance of the "pseudo-21 helical screw symmetry". In general, the manuscript was very difficult to read for an audience with limited expertise in SSNMR. This reduces the ability of the findings to reach a wider audience and does not allow the authors to get their message across.

Can the authors discuss the significance of the anti-parallel structures found in the LBD filaments? Typically this arrangement results in lower stability and reduced efficiency in elongation and seeding.

For figure 6, I assume the data points were fibrils extracted from different LBD and MSA cases, however, this is not explicitly stated in the legend. Also, were the differences statistically significant? The data seem to not fit a normal distribution- I recommend consultation with a statistician.

The extracts from DLB were acquired from postmortem tissues. Is it possible that changes in pH, salt in post mortem tissue could change the structure of the filaments so they do not resemble those in live tissue.

Reviewer #2:

Remarks to the Author:

The manuscript by Dhavale et al. describes two major achievements/advances as stated by the authors; 1) Development and validation of a novel method to amplify A β fibrils extracted from LBD postmortem tissue samples; and 2) determination of the structure of these amplified fibrils at an atomic resolution using SSNMR. This reviewer is not an expert in SSNMR. Therefore, this review will focus primarily on the first claim.

This new method is based on six cycles of amplification of brain-derived aggregates. Unfortunately, the authors do not provide all the necessary data to assess 1) the robustness and reproducibility of this protocol, 2) the structural and morphological heterogeneity of the amplified fibril preparations, 3) the ability of these fibrils to induce the formation of distinct aSyn aggregates or pathologies in cellular models of aSyn fibrillization and pathology formation. Therefore, more data is needed to evaluate their claims and establish that the amplified structures they study by SSNMR reflect the structure of the brain-derived fibrils rather than the in vitro aggregation and amplification conditions.

1. The rationale for applying six cycles of amplification is not clear. Why not 3 or 4?
2. The authors should present a more detailed characterization of the fibrils generated at each cycle, including
 - a. EM data that capture the morphological homogeneity/heterogeneity of the fibrils at each stage, EM images of larger fields with multiple fibrils and not just one isolated fibril, i.e., accurate representation of the different morphologies in the sample
 - b. Quantitative assessment of their dimension (e.g., diameter)
 - c. Proteinase K (PK)-resistant profile. This is essential to 1) show that the fibrils exhibit a distinct PK profile/conformation from the unseeded samples or samples seeded with recombinant proteins; 2) demonstrate that the structure and morphology of the amplified fibrils are replicated from one cycle to another.
 - d. Purity of the fibril preparations after each cycle, by SDS-PAGE analysis, not WB. This is essential to rule out sample degradation during the amplification cycles.
3. Given that the total volume in the aggregation vessel changes from one cycle to another. This changes the air-water interface, which has been shown to influence the aggregation properties of

aSyn and other amyloid proteins. This is another reason why the nature of the aggregates at each cycle should be characterized.

4. The author should establish the robustness and reproducibility of their protocol using the methods outlined above (TEM, PK-resistant profile, and ssNMR) and using multiple samples from the same LBD sample preparation and using multiple LBD samples.

5. Page 4, the authors indicate, "Negative-stain transmission EM analysis of amplified LBD fibrils showed straight fibrils with a diameter of 10-18 nm and no visible twist (Fig. 1b,d). Fibrils amplified from control tissue samples had smaller diameters and more curvature (Extended Data Fig. 1a,b)."

a. The original EM images and not selected fibrils should be shown

b. A direct comparison of the fibrils amplified from LBD and Control samples should be presented in the main figure.

c. The PK-resistance profile of both preparations, EM and SDS-PAGE analysis of these samples should be presented.

d. The aggregation kinetics profiles of alpha-synuclein monomers seeded with LBD or control samples should be presented. This is important to determine the role of de novo aggregation and/or differences in seeding efficiency between the two preparations.

6. The HEK293 seeding studies are not informative. Although they showed differences in the appearance of the newly seeded aggregates, they do not indicate to what extent the amplified fibrils from LBD or MSA tissues alter the structural or biochemical properties of the newly formed aggregates, compared to samples from controlled tissues or recombinant fibrils. Furthermore, the quality control data are missing; 1) evidence that the same amounts of aggregate seeds was used; 2) comparison of alpha-synuclein levels in both preparations; 3) quality control of the amplified fibril preparations, EM, quantitative analysis of their size, diameter, and length. Finally, the authors speculated on why the soluble fraction of the MSA sample induces alpha-synuclein aggregation in HEK293 cells but did not provide any experimental data to support their hypotheses. Analysis of the aggregation state of these samples by EM and WB could have provided some insights and ruled out the presence of fibrils in these samples.

7. The experiments on fibril growth are not informative unless the authors can correlate differences in growth rates to differences in fibril structure, either directly by SSNMR or indirectly by assessing the biochemical and PK-resistant profiles of the fibrils.

8. The authors' specular extraction with extraction with sarkosyl converts two protofilament fibrils to single protofilament fibrils". It is unclear why they did not test this experimentally since they showed that the amplified fibrils are composed of two protofilaments. Does treatment of these two protofilament fibrils lead to their conversion to single protofilament fibrils?

9. What is the % of alpha-synuclein monomers that go through the 50 KDa filter?

Reviewer #3:

Remarks to the Author:

This paper reports the use of solid-phase NMR to examine the core structure of Asyn fibrils derived from LBS postmortem tissue samples. The resulting fold is largely in agreement with a recent Cryo-EM structure of a minority population of twisted single protofilament fibrils extracted from LBD tissue. Mutational studies confirm the validity of the resulting structure.

This study gives important new insight into the formation of AS fibrils using different preparative routes and merits publication in Nat Com. However, it would be useful to be more specific on experimental details and analysis as well as include previous work as follows:

- Page 5 suggests that the ssNMR analysis is entirely based on using uniformly ^{13}C , ^{15}N and ^{13}C , ^2H , ^{15}N labeled LBD fibrils. However, the methods and supplementary section mention preparation of and ssNMR data of 2-glycerol labelled samples. It may help to include a comprehensive table with samples used, experiments performed and restraints retrieved.

- Page 5 reports "very sharp" peaks: It would be good to quantify line width, in particular in the ¹H dimension (vide infra) and report whether the line width varies between core and "hinge" residues incl. amino acids found in the vicinity of the N or C termini. Also, for many correlations in Figure 3A the resolution is limited and fig 3b may be indicative of peak doubling (e.g., G41). Can the authors exclude the presence of a 2nd population in their data? Please comment.

- Page 5 concludes that "This spectrum suggests a highly ordered fibril core and a disordered N- and C terminus, consistent with previous studies of fibrils and of A_β in particular": Wouldn't ssNMR be particularly useful to study these segments in further detail? In any case the authors should cite appropriate references regarding "previous studies" including recent work by Zhang S et al (JACS 2023).

- They also mention the clever use 2H₂O/1H₂O treatment to simplify the analysis to the rigid fibril core. However, it would be good to clarify what the authors define a "fibril core": for example- is it defined by the heat maps shown in Figure 7d/e?

- Figure S3 gives a very useful overview of the NMR analysis. Couldn't the lack of aa assignments be related to either limited exchange or conformational dynamics? Hence, a discussion on preparative artifacts or intrinsic fibril properties should be included - for example on page 9 regarding the ssNMR data. Did the authors consider complementary methods such as Mass spectrometry to characterize the nature of their preparations?

- Figure 3B contains 3D CO-N-H and CA-N-H data which were recorded at comparatively low MAS rates. It would be useful to include in the SI projection along the NH plane. Also, shouldn't the axis labelling in Figure 3b be f2 for the ¹⁵N plane and f1 for the ¹³C plane?

- Structure calculation: Several long-range correlations shown in Figure 4a seem to exhibit limited sensitivity and/or are located in overlapping spectral regions. It may help to include a table with the final set of restraints per residue, in particular those that were added manually as described in the main text.

-Lastly, seeding has been used in other ssNMR studies on amyloid fibrils. Please cite the relevant literature when discussing the "novel" method (page 3).

Response to Reviewers' Comments

We appreciate all of the constructive comments from the reviewers.

In response to the reviewers' comments, we substantially revised the manuscript with new data to further characterize structural features of amplified fibrils, including new data that further demonstrates consistency and reproducibility of the LBD amplified fibril preparations. We added new SSNMR data from two additional LBD cases that demonstrates consistent structural features across multiple LBD cases. Furthermore, we added substantial data to characterize additional structural features across multiple LBD cases, utilizing methods that include EM, protease digestion, and seeding properties in a biosensor cell line. Finally, we have emphasized that multiple independent amplified fibril preparations from the same case have consistent SSNMR structural features.

We revised the manuscript to address each of the reviewers' comments as outlined below.

Reviewer 1

Reviewer Comment 1: The biggest concern is that the structure appears to be derived from only one LBD case. There also appear to be no statistics. How can the authors be sure these findings are consistent across multiple cases and this was just an artifact from one brain with a relatively long postmortem interval?

Response: We now include analysis of 1D SSNMR spectra from two additional cases and 2D SSNMR spectra from one additional case (Results section, page 9, lines 371-381 and Extended Data Figure 9) and show that the spectra are highly similar to that of the case used for full structure determination, indicating a high degree of structural similarity in their fibril cores. As we point out in the Discussion section, some amount of structural variation between cases may exist and this will be a focus of future studies. However, our analysis indicates a high degree of similarity, similar to the observations by Yang et al. We included postmortem intervals for the list of cases utilized for structural studies in Supplementary Table 8 (Supplementary Notes, page 19, line 283).

Reviewer Comment 2: What were the brain regions used for control, LBD and MSA cases? It is likely that different conformations exist across different areas of the brain.

Response: We listed the brain regions used for all cases in Supplementary Table 8 (Supplementary Notes, page 19). Relevant to this point, the two additional cases we analyzed by comparison of 1D and 2D SSNMR spectra were from a cortical region (middle frontal gyrus) and the case for full structure determination was from caudate, which indicates a high degree of similarity between cortical and basal ganglia regions (Results section, page 9, lines 371-382).

Reviewer Comment: It is also possible that one form of filament amplifies more than others because it is more concentrated and the amplification reaction favors this more stable, dominant form. It is also possible that important assemblies are lost in Tx100 fraction and that focusing on the insoluble fraction selects for only one type of LBD filament. These are important caveats for this and previous studies using amplification methods.

Response: We agree and highlight these points in the Discussion section, paragraph 2. Importantly, we also revised the text to make it clearer that our amplification protocol produces a combination of single and two protofilament forms with the same fold. We previously described a combination of single and two protofilament fibrils in the results of 2D classification for single

particle cryo-EM data (Results section, page 6, lines 191-201), but we now highlight the combination of single and two protofilament fibrils in multiple places in the manuscript, including the abstract. The SSNMR model was developed specifically for the two protofilament form in order to understand the interface for that form, but the single set of peaks in the NMR data indicate that the single protofilament form has the same fold. We added more discussion (Discussion Section, page 11, lines 460- 472) that the relative amounts of the two forms may be determined by extraction and amplification conditions, and also the possibility that amplification conditions may promote the formation of two protofilament fibrils. Future studies will help to understand these questions.

Reviewer Comment 3: The biosensor cells with high levels of expressed A53T-a-synuclein are okay. However, there is no quantitation of the percentage of cells with aggregates. Also, it would have a greater impact on the field to see seeding in primary neurons which have high concentrations of normal synuclein at synapses and do not need a lipid carrier for entry of the fibrils into the neurons. Alpha-Synuclein expressed in immortalized cells is not converted to an aggregated form as easily as alpha-synuclein in neurons. The “soluble” DLB filaments could seed alpha-synuclein aggregates in primary neurons.

Also, while not necessary for this study, it is important to note that soluble alpha-synuclein assemblies could show different morphology compared to the insoluble seeds. Such data would also be useful to the field for determining the impact of DLB assemblies on neuronal function and spread in the nervous system.

Response: We now provide quantification of inclusions formed after seeding the biosensor cells with LBD and MSA amplified fibrils in Supplementary Table 7 (Supplementary Notes, page 18, line 279). We utilized the biosensor cell line based on our previous studies demonstrating distinct seeding properties for LBD and MSA tissue fractions, including differences in seeding efficiency and inclusion morphology. We observe that the amplified fibrils have distinct seeding properties that are similar to those observed for tissue fractions, providing one approach to guide the development of the amplification protocol. We agree that seeding experiments in neurons could provide additional insight about mechanisms and are pursuing this approach as a future direction.

Reviewer Comment 4: Furthermore, according to the methods “For LBD amplified fibrils samples, concentrations between 26 nM to 1 nM were used. For MSA amplified fibril samples, concentrations between 1.5-0.1 nM was used.” Ideally a concentration curve should be performed with quantitation of aggregates quantified at the different concentrations.

Response: The primary goal of the experiments in the biosensor cell line was to provide an assessment of morphology of aggregates produced by LBD and MSA amplified fibrils. We now also provide quantification of aggregates for LBD and MSA amplified fibrils in Supplementary Notes Table 7 (Supplementary Notes, page 18, line 279). This quantitative analysis of aggregates in relation to concentration of fibril seeds demonstrates substantially higher seeding efficiency for MSA fibrils. We similarly found that MSA tissue fractions had substantially higher seeding efficiency compared to LBD tissue fractions in our previous study, which provides the foundation for interpreting these results (Results section, page 5, lines 158-162). The combination of morphology assessment and seeding efficiency measurements in the biosensor cells demonstrates distinct properties of the amplified fibril conformers that are similar to the properties of the tissue fractions, which provides a guide for amplification.

Reviewer Comment 5: The BCA assay is an indirect and not correct method of measuring the synuclein concentration. The fibrils can be dissociated into monomer with guanidine hydrochloride and the protein then either measure using BCA or ideally, using Beer-Lambert law.

Response: We used our radioligand assay to determine concentration of amplified fibrils used in the biosensor cell experiments and to guide development of the amplification method. In addition to high sensitivity, the radioligand assay provides high specificity for fibrils over monomer, eliminating the issue with monomer carryover when fibrils are isolated by sedimentation and measured in BCA or A_{280} assays. In response to the reviewer's comment, we used A_{280} measurement of guanidine-treated fibrils to further calibrate the radioligand assay, for more accurate conversion of radioligand binding measurements to fibril concentration. We also validated the radioligand measurements by performing additional A_{280} measurements of isolated fibrils for several amplified fibril preparations (Results section, page 4, lines 120-122 and Supplementary Tables 4, 5, 6).

Reviewer Comment 6: Also, unlike for the insoluble form of synuclein, for soluble tissue fraction, Tx-100 was not used and thus some assemblies likely were not extracted and lost to the pellet. It is convincing (although quantitation would help solidify these data) that the "soluble" fraction from MSA seeds aggregates in the biosensor cells, and the LBD "soluble" fraction does not. It is possible some assemblies would be available in the Tx-100 extracted "soluble" fraction—this fraction could be dialyzed to remove triton. While not necessary for this paper, it would be interesting to know if some "Tx-100" soluble synuclein assemblies have seeding activity.

Response: We agree this would be interesting for future studies of seeding activity in soluble fractions.

Reviewer Comment 7: For figure 1d, did the authors quantify the percentage of fibrils with paired helical filaments and straight fibrils across all preparations from multiple fields? It is difficult to make conclusions from EM snapshots.

Response: We added quantification of diameter measurements obtained from multiple TEM images in Supplementary Notes Fig. 7-8. However, we did not quantify helical, paired helical or straight appearance since we were not confident that they can be reliably distinguished in negative stain TEM images.

Reviewer Comment 8: Were the cases used pathologically confirmed to show Lewy pathology, neurofibrillary tangles, amyloid beta? Were any of the cases genotyped for ApoE4?

Response: We added information on the histopathology for each autopsy case to Supplementary Table 8 (Supplementary Notes, page 19, line 283). We do not have ApoE4 genotype available.

Reviewer Comment 9: Can the authors discuss the significance of the "pseudo-21 helical screw symmetry". Can the authors discuss the significance of the anti-parallel structures found in the LBD filaments? Typically this arrangement results in lower stability and reduced efficiency in elongation and seeding.

Response: We added a sentence about this in the Discussion section (Page 12, lines 544 - 547) and agree this is an interesting question and a future direction for further investigation.

Reviewer Comment 10: In general, the manuscript was very difficult to read for an audience with limited expertise in SSNMR. This reduces the ability of the findings to reach a wider audience and does not allow the authors to get their message across.

Response: In this revised manuscript, we have worked to clarify the implications of the SSNMR data including the interpretation of chemical shift patterns in terms of secondary structure, and we have moved additional technical details to the Supplementary Notes section. We have simplified language and added clarifying statements throughout the SSNMR discussion in order to translate the findings into tone more suitable for a general structural biology audience.

Reviewer Comment 11: For figure 6, I assume the data points were fibrils extracted from different LBD and MSA cases, however, this is not explicitly stated in the legend. Also, were the differences statistically significant? The data seem to not fit a normal distribution- I recommend consultation with a statistician.

Response: We clarified in the legend for Figure 6 that data points represent different cases (Page 31, lines 1098-1113). We also added statistics to address the primary aim of this experiment, which was to determine whether growth rates of the LBD amplified fibrils were significantly impaired for each of the mutant monomers relative to WT monomer. The LBD data appears to have a normal distribution (indicated by Shapiro-Wilk test) and we used unpaired, two-tailed t-tests (Welch) and corrected for multiple comparisons with the Holm-Bonferroni method. The MSA amplified fibrils had higher variability in this experiment as the reviewer noted and the differences between mutant and WT monomer were not significant for the MSA fibrils with either nonparametric or parametric statistics.

Reviewer 2

Reviewer Comment: Unfortunately, the authors do not provide all the necessary data to assess 1) the robustness and reproducibility of this protocol, 2) the structural and morphological heterogeneity of the amplified fibril preparations, 3) the ability of these fibrils to induce the formation of distinct aSyn aggregates or pathologies in cellular models of aSyn fibrillization and pathology formation. Therefore, more data is needed to evaluate their claims and establish that the amplified structures they study by SSNMR reflect the structure of the brain-derived fibrils rather than the in vitro aggregation and amplification conditions.

Response: We have carefully considered concerns about specificity and reproducibility of the amplification protocol and added substantial further characterization to support the robustness and reproducibility of the protocol, as well as additional data that assesses the structural and morphological heterogeneity of the amplified fibrils. We outline details in response to individual comments below.

Reviewer Comment 1: The rationale for applying six cycles of amplification is not clear. Why not 3 or 4?

Response: The number of cycles was determined based on the mass of fibrils needed for structural studies. We now state this in the Results section (Page 6, lines 205-206). Six or more cycle were required to generate sufficient material for SSNMR studies, so we utilized fibrils amplified for 6 cycles for the majority of our characterization.

Reviewer Comment 2: The authors should present a more detailed characterization of the fibrils generated at each cycle, including

- a. EM data that capture the morphological homogeneity/heterogeneity of the fibrils at each stage, EM images of larger fields with multiple fibrils and not just one isolated fibril, i.e., accurate representation of the different morphologies in the sample
- b. Quantitative assessment of their dimension (e.g., diameter)
- c. Proteinase K (PK)-resistant profile. This is essential to 1) show that the fibrils exhibit a distinct PK profile/conformation from the unseeded samples or samples seeded with recombinant proteins; 2) demonstrate that the structure and morphology of the amplified fibrils are replicated from one cycle to another.
- d. Purity of the fibril preparations after each cycle, by SDS-PAGE analysis, not WB. This is essential to rule out sample degradation during the amplification cycles.

Response: We added data as follows:

- a. We added additional EM images to Supplementary Notes Fig. 1-6 (Supplementary Notes, pages 5-10), which include six EM micrographs each for LBD amplified fibrils after 2, 4 and 6 cycles, as well as MSA and control amplified fibrils after 6 cycles.
- b. We added quantitative analysis of fibril diameter in Supplementary Notes Fig. 7-8 (Supplementary Notes, pages 11-12).
- c. We added PK digestion profiles for amplified fibrils obtained after 6 cycles of amplification, in Supplementary Notes Fig. 9 (Supplementary Notes, page 13). We did not add PK digestion profiles for samples with fewer amplification cycles since we saw only minor differences between LBD, MSA and control amplified fibril preparations at 6 cycles of amplification and therefore digestion profiles after fewer cycles would not provide further information about structural variation with respect to amplification cycles.
- d. We provide SDS-PAGE analysis in Supplementary Notes Fig. 9 (Supplementary Notes, page 13). In early cycles of amplification, we do observe limited proteolytic cleavage of A β , which is likely related to proteases present in insoluble fractions from brain tissue. Similar proteolytic fragments are also observed for A β extracted from human brain tissue.

All of the above new data providing further characterization is presented in a new subsection within the Results section (Pages 5-6, lines 165-190).

Reviewer Comment 3: Given that the total volume in the aggregation vessel changes from one cycle to another. This changes the air-water interface, which has been shown to influence the aggregation properties of A β and other amyloid proteins. This is another reason why the nature of the aggregates at each cycle should be characterized.

Response: We added the additional characterization as outlined above in response to Comment 2 from Reviewer 2.

Reviewer Comment 4: The author should establish the robustness and reproducibility of their protocol using the methods outlined above (TEM, PK-resistant profile, and ssNMR) and using multiple samples from the same LBD sample preparation and using multiple LBD samples.

Response: We addressed this with amplified fibril samples from multiple cases in the additional characterization outlined in our response to Comment 2, where we used at least 3 LBD cases. We also emphasize that we have analyzed NMR data for multiple independently prepared amplified fibril samples derived from the same case but produced with different isotopic labeling

strategies, and we see nearly identical spectra as we state in the Results section (page 6, lines 203-207), indicating a high degree of consistency in the amplification protocol.

Reviewer Comment 5: Page 4, the authors indicate, “Negative-stain transmission EM analysis of amplified LBD fibrils showed straight fibrils with a diameter of 10-18 nm and no visible twist (Fig. 1b,d). Fibrils amplified from control tissue samples had smaller diameters and more curvature (Extended Data Fig. 1a,b).”

- a. The original EM images and not selected fibrils should be shown.
- b. A direct comparison of the fibrils amplified from LBD and Control samples should be presented in the main figure.
- c. The PK-resistance profile of both preparations, EM and SDS-PAGE analysis of these samples should be presented.
- d. The aggregation kinetics profiles of alpha-synuclein monomers seeded with LBD or control samples should be presented. This is important to determine the role of de novo aggregation and/or differences in seeding efficiency between the two preparations.

Response:

- a, b. We revised figure 1 (Page 26) to include more EM images for LBD, MSA and control fibrils. We also added a substantial number of additional EM images to Supplementary Notes Figures 1-6 (Supplementary Notes, pages 5-10).
- c. We provided SDS-PAGE analysis of 6th cycle amplified fibrils with and without PK digestion in Supplementary Notes Fig. 9 (Supplementary Notes, page 13). We provided quantitative analysis of fibril diameter in Supplementary Notes Fig. 7-8 (Supplementary Notes, pages 11-12).
- d. Kinetic data is shown in the Supplementary Notes Table 1 (Supplementary Notes, page 17), which indicates that low levels of fibrils accumulate in control samples, although it is unclear whether this is due to de novo nucleation of monomer or the presence of Asyn fibril seeds in control tissue samples from older individuals. We favor de novo nucleation induced either by sonication or by interaction of monomer with tissue components.

Reviewer Comment 6: The HEK293 seeding studies are not information. Although they showed differences in the appearance of the newly seeded aggregates, they do not indicate to what extent the amplified fibrils from LBD or MSA tissues alter the structural or biochemical properties of the newly formed aggregates, compared to samples from controlled tissues or recombinant fibrils. Furthermore, the quality control data are missing; 1) evidence that the same amounts of aggregate seeds was used; 2) comparison of alpha- synuclein levels in both preparations; 3) quality control of the amplified fibril preparations, EM, quantitative analysis of their size, diameter, and length. Finally, the authors speculated on why the soluble fraction of the MSA sample induces alpha-synuclein aggregation in HEK293 cells but did not provide any experimental data to support their hypotheses. Analysis of the aggregation state of these samples by EM and WB could have provided some insights and ruled out the presence of fibrils in these samples.

Response: Experiments examining amplified fibril preparations in the 293 biosensor cells were primarily designed to compare inclusion morphology as an approach to monitor structural properties of amplified fibrils, based on previously observed differences in seeding for fibrils isolated from LBD and MSA tissue (Yamasaki et al, JBC 2019). However, we now also include quantitative analysis of seeding efficiency demonstrating substantially higher seeding efficiency

for MSA amplified fibrils, consistent with the properties observed for fibrils isolated from tissue (Results section, page 5 lines 158-160 and Supplementary Table 7). We also now include detailed characterization of the amplified fibril preparations as outlined in our responses to Comment 2 from Reviewer 2 above, providing the quality control data as recommended. We also address precision of amplified fibril concentration measurements used as seeds in our response to Reviewer 1 Comment 5. Finally, we analyzed the soluble fractions by negative stain EM but were not able to reliably identify oligomeric or short fibrillar species. This may be due to limited sensitivity of the negative stain EM for capturing and visualizing fibrils on grids. We do not speculate on the structural basis of seeding activity in the MSA soluble fraction, a question that can be pursued in future studies, but utilize this observation to indicate that seeding properties of amplified fibrils are similar to those of the tissue fractions.

Reviewer Comment 7: The experiments on fibril growth are not informative unless the authors can correlate differences in growth rates to differences in fibril structure, either directly by SSNMR or indirectly by assessing the biochemical and PK-resistant profiles of the fibrils.

Response: The fibril growth rate assays add more information on structural properties of LBD fibrils by determining the roles of specific amino acid residues in LBD fibril growth. Comparison to MSA and IV_{Tris} fibril conformers is included for additional information. Our data comparing structural properties of fibrils now include differences in morphology (EM), PK-digestion profiles, and seeding properties in a biosensor cell line for LBD and MSA fibrils. Our previously published data on SSNMR analysis of in vitro assembled fibrils (IV_{Tris}) also demonstrate structural features distinct from amplified LBD fibrils (Barclay et al. BNMR 2018).

Reviewer Comment 8: The authors' speculate that extraction with sarkosyl converts two protofilament fibrils to single protofilament fibrils". It is unclear why they did not test this experimentally since they showed that the amplified fibrils are composed of two protofilaments. Does treatment of these two protofilament fibrils lead to their conversion to single protofilament fibrils?

Response: We agree that treating the amplified fibrils with sarkosyl may be one approach to understand whether extraction conditions alter the distribution between single and two protofilament fibrils. In response to this comment, we analyzed fibril diameter after treatment with 2% sarkosyl and the results did not indicate a change in diameter (Results section, page 5, lines 176-179). One limitation is that exposure of amplified fibrils to 2% sarkosyl, even in the context of a protocol that simulates the tissue extraction protocol, may not fully replicate the process by which A β fibrils are extracted from postmortem brain tissue samples. We modified the Discussion section (Page 11, lines 453-470) to highlight that extraction is less likely to account for the lack of two protofilament fibrils observed in the cryo-EM studies and discuss other potential explanations, including the possibility that two protofilament fibrils may arise or increase in number due to fibril-fibril association during the amplification process, as suggested by quantitative analysis of fibril diameters.

Reviewer Comment 9: What is the % of alpha-synuclein monomers that go through the 50 KDa filter?

Response: Approximately 70%. This information was added to the Methods section (Methods, page 14, lines 579-580).

Reviewer #3 (Remarks to the Author):

This paper reports the use of solid-phase NMR to examine the core structure of Asyn fibrils derived from LBS postmortem tissue samples. The resulting fold is largely in agreement with a recent Cryo-EM structure of a minority population of twisted single protofilament fibrils extracted from LBD tissue. Mutational studies confirm the validity of the resulting structure.

This study gives important new insight into the formation of AS fibrils using different preparative routes and merits publication in Nat Com. However, it would be useful to be more specific on experimental details and analysis as well as include previous work as follows:

Reviewer Comment 1. Page 5 suggests that the ssNMR analysis is entirely based on using uniformly ^{13}C , ^{15}N and ^{13}C , ^2H , ^{15}N labeled LBD fibrils. However, the methods and supplementary section mention preparation of and ssNMR data of 2-glycerol labelled samples. It may help to include a comprehensive table with samples used, experiments performed and restraints retrieved.

Response: We updated Supplementary Table 9 (Supplementary Notes, page 20) to include the ^{13}C glycerol sample in the list of SSNMR experiments, which provides details for the experiment with this sample. We describe the use of that sample for additional distance restraints in the “Distance restraints from solid-state NMR” section of Supplementary Notes (Supplementary Notes, page 2, line 59). We also added a figure describing the manual, automated and total restraints for each residue in the structure in Supplementary Notes Fig. 12. (Supplementary Notes, page 16), as well as a final refinement statistics table, describing the final restraints and violations of our structure in Supplementary Table 12 (Supplementary Notes, page 23).

Reviewer Comment 2. Page 5 reports “very sharp” peaks: It would be good to quantify line width, in particular in the ^1H dimension (vide infra) and report whether the line width varies between core and “hinge” residues incl. amino acids found in the vicinity of the N or C termini. Also, for many correlations in Figure 3a the resolution is limited and fig 3b may be indicative of peak doubling (e.g., G41). Can the authors exclude the presence of a 2nd population in their data? Please comment.

Response: We have changed this statement to “sharp peaks” to moderate the language (Page 6, lines 207-209), and more importantly we have quantified the linewidths of the ^{13}C , ^{15}N and ^1H peaks and added graphs of linewidths for each residue number in Supplementary Notes Fig. 10 (Supplementary Notes, page 21). Regarding the possibility of peak doubling at G41, we have assigned that peak as the G41 to Y39 correlation in the hCONH 3D. This region has exceptional signal to noise so the i to $i-2$ correlation is present and not a result of peak doubling. We have chosen to preserve the figure format in order to emphasize the completeness of assignments for the observed peaks.

Reviewer Comment 3. Page 5 concludes that “This spectrum suggests a highly ordered fibril core and a disordered N- and C terminus, consistent with previous studies of fibrils and of Asyn in particular”: Wouldn’t ssNMR be particular useful to study these segments in further detail? In any case the authors should cite appropriate references regarding “previous studies” including recent work by Zhang S et al (JACS 2023).

Response: We thank the reviewer for this insightful comment that highlights the strengths that SSNMR has to investigate the ordered core and the disordered termini. We plan future studies to assign the termini in greater detail, following the example from the Zhang publication, which is now cited in the manuscript also. We included a citation for the recent work from Zhang S et al (References, page 35, line 1254).

Reviewer Comment 4. They also mention the clever use 2H₂O/1H₂O treatment to simplify the analysis to the rigid fibril core. However, it would be good to clarify what the authors define a “fibril core”: for example- is it defined by the heat maps shown in Figure 7d/e?

Response: For purposes of this experiment, the core is defined as the region resistant to amide proton exchange. This corresponds to residues 34 – 45 and 63 – 95. We edited the text in the Results section to clarify this as pasted below (Results section, page 7, lines 251-254) :

“The data provided by the ¹³C and ¹H detection experiments were leveraged to yield dihedral restraints (Fig. 3d) and predicted random coil index order parameter (RCI S²), revealing LBD fibrils form highly ordered beta-sheet fibril core structure involving E34 to K45 and V63 to V95.”

Reviewer Comment 5. Figure S3 gives a very useful overview of the NMR analysis. Couldn't the lack of aa assignments be related to either limited exchange or conformational dynamics? Hence, a discussion on preparative artifacts or intrinsic fibril properties should be included - for example on page 9 regarding the ssNMR data. Did the authors consider complementary methods such as Mass spectrometry to characterize the nature of their preparations?

Response: We agree that the lack of signals of the N- and C- termini is attributable to a combination of static and dynamic disorder. We have added additional details to the discussion section to emphasize this point. In addition, we prepared the MALDI figure (Supplementary Notes Fig 11, page 15) that characterizes the isotopic enrichment of Asyn monomers used for the SSNMR sample preparations:

Reviewer Comment 6. Figure 3B contains 3D CO-N-H and CA-N-H data which were recorded at comparatively low MAS rates. It would be useful to include in the SI projection along the NH plane. Also, shouldn't the axis labelling in Figure 3b be f2 for the 15N plane and f1 for the 13C plane?

Response: We added the 1H-15N 2D projection in Supplementary Notes Fig. 10 (Supplementary Notes, page 14) which is also included above in our response to Reviewer Comment 2. We corrected the axis labeling in Figure 3b and thank the reviewer for noting this (Figures, page 28).

Reviewer Comment 7. Structure calculation: Several long-range correlations shown in Figure 4a seem to exhibit limited sensitivity and/or are located in overlapping spectral regions. It may help to include a table with the final set of restraints per residue, in particular those that were added manually as described in the main text.

Response: We added a figure showing the restraints for the structure in Supplementary Notes Fig. 12 (Supplementary Notes, page 16), as we also highlight in our response to Reviewer Comment 1.

Reviewer Comment 8. Lastly, seeding has been used in other ssNMR studies on amyloid fibrils. Please cite the relevant literature when discussing the "novel" method (page 3).

Response: We edited the discussion section to highlight that our amplification method is based in part on previous studies utilizing seeded fibril growth and added several citations (Discussion section, page 11-12, lines 483, 515-520).

Thank you for considering our revised manuscript for publication in *Nature Communications*.

Sincerely,

Chad Rienstra PhD
Professor of Biochemistry
University of Wisconsin – Madison
crienstra@wisc.edu

Paul Kotzbauer MD, PhD
Professor of Neurology
Washington University School of Medicine
kotzbauerp@neuro.wustl.edu

Reviewers' Comments:

Reviewer #1:

Remarks to the Author:

For me the inclusion of 1D SSNMR spectra from two additional cases and 2D from one additional case greatly strengthens the study because the replicability indicates the structures of the fibril core is generalizable to DLB. In addition, amplification of fibrils from DLB will likely result in similar structures regardless of the source of DLB brain tissue and thus others in the field can replicate the experiments.

The study also replicates the structure of the single protofilament using cryo-EM from Yang et al. and extends the findings to show that amplified DLB fibrils are a mixture of a single protofilament and two protofilament fibrils with a low twist.

This is a very important study for the field.

Reviewer #2:

Remarks to the Author:

1. One of my MAJOR/ MAIN Concerns with this work is highlighted in the data the authors now provide in Supplementary Note Figure 9. It shows that the samples show aSyn truncations that are seen to be present in the undigested 2nd Cycle samples and that increase significantly in the 4th Cycle undigested samples. In fact, for some samples, the amount of intact full-length aSyn represents only a minor species, e.g. 4th Cycle samples (LB1, LBD2, MSA2, and MSA3). Even in the other samples the ratio of ~50/50 of full-length to truncated samples. This raises several important questions about what is being amplified and whether mixtures of full-length and truncated amplified fibrils are formed. This has significant implications for the interpretation of the data, see below.

2. Panel C in Supp Notes Fig 9 show several inconsistencies and the WB does not suggest that these sample represent aggregated/amplified samples for the following reasons; 1) all the amplified samples from the 2nd and 4th Cycle without exception show the presence of high molecular weight species and band streaks, reflecting the presence of aggregated/fibrillar forms of aSyn; and 2) the presence of truncated species in significant amounts. None of these features is shown in the samples from the 6th amplification cycle, suggesting that the samples used here reflect primary soluble and not aggregated forms of aSyn. Consistent with this is the fact that all samples, including the controls, show a similar PK digestion band pattern, which is not expected, especially between LBD1 and MSA cases (Figure 1b and 1d). The SDS-PAGE profiles of the different cycles show that the fibrils obtained in each cycle have different biochemical compositions. Finally, the presence of truncated aSyn and these large variations from one cycle to another suggests that the procedure is not reproducible.

Supp Note Figure 9 MUST BE INCLUDED IN THE MAIN FIGURES of the revised version of the manuscript. Without this data, it is not possible for the reader to accurately interpret the results.

3. The band intensity for the amplified samples (Supp notes figure 6, Cycle 6) suggests high variability in the concentration of aSyn among all the samples, which could also influence the in vitro kinetics and cell seeding studies.

4. Biosensor assays: The presence of distinct and variable amounts of full-length and truncated fibrils makes it difficult to interpret the seeding results using the biosensor assay. Are the seeding or morphological differences due to the different structure of the amplified full-length, truncated aSyn, or a combination of both? Difficult to answer. Furthermore, at the level of characterization provided, it is not possible to make any statement about similarities or differences in the morphological properties of the newly seeded aggregates. If they want to expand on this, please

highlight these differences more clearly. They all appear filamentous. No quantification is shown in the main figure, plotted. This is important to allow a direct comparison of the seeding and biochemical data for each sample, WB.

5. The authors should assess the amplified sample from the MSA soluble fraction by WB to see if it contains truncated aSyn species or quantify the amount of soluble and aggregated aSyn species. This could provide some insight as to why they see seeding with the soluble MSA fraction.

6. Fibril growth experiments. Again, the interpretation of the kinetic data and growth rates is not complicated by the variable heterogeneity of the samples due to the presence of truncated forms of aSyn.

7. In the revised version, no PK digestion data were included as requested. Only data for the samples from Cycle 6 for which this reviewer questions whether they are indeed amplified samples and no differences are seen among the LBD, MSA, and control samples that reflect structural differences.

8. "We developed a new method to amplify aSyn fibrils from LBD postmortem tissue, enabling high-resolution analysis of fibril structure. 2D classification results from single-particle cryo-EM indicate a combination of single protofilament and two protofilament fibrils with very low twist." 2D class averages and evidence for double filaments are only shown for LBD, no data were shown for MSA and control samples. This is important since their findings differ from those of a recent study showing primarily single protofilament fibrils. Could this be due to the presence of fibrils derived from full-length and truncated aSyn protein?

1. All of the above does not support one of the authors' main claims that they "We developed a new method to amplify aSyn fibrils from LBD postmortem tissue, enabling high resolution analysis of fibril structure."

Minor Comments

1. The authors should provide a more descriptive analysis of the differences in the morphological properties and diameters of the fibrils obtained in the different cycles. For example, 1) Supp Notes figure 3 compared to 1,2, 4, 5, 6, and 2) difference in diameter between amplification cycles 2 and 4 and amplification cycle 6. The data suggest the appearance of a second population of fibrils with diameters of ~13 nm.

For example, what is meant by distinct here "These fibrils appear distinct from all forms of amplified fibrils, based on diameter and twist"

Very vague interpretation of the data. What information could be derived about each population, e.g., size to morphology correlation

"Diameter measurements for LBD amplified fibrils indicated a bimodal distribution after 6 cycles of amplification, with diameters ranging from 6 nm to 15 nm, suggesting two populations with overlapping diameter measurements. A bimodal distribution was not apparent for fibrils obtained

Reviewer #3:

Remarks to the Author:

The authors have appropriately addressed the issues raised earlier and provided important additional data and information in the revised version.

Response to Reviewers' Comments

We appreciate the comments from the reviewers.

In this response, we further revised the manuscript with additional data and text to address questions about the heterogeneity of fibril composition based on the presence of C-terminal truncations in amplified A β fibrils. We include additional data (western blot) to further characterize these cleavage products, demonstrating that these small (2-3 kD) cleavages occur at the C-terminus, and therefore are highly similar to those found in A β fibrils isolated from human Lewy body dementia postmortem tissue. We also clarified that these cleavages occur within the disordered C-terminal region of fibrils and therefore are unlikely to affect the beta sheet fibril structure characterized in cryo-EM and SSNMR studies.

Reviewer #1 (Remarks to the Author):

For me the inclusion of 1D SSNMR spectra from two additional cases and 2D from one additional case greatly strengthens the study because the replicability indicates the structures of the fibril core is generalizable to DLB. In addition, amplification of fibrils from DLB will likely result in similar structures regardless of the source of DLB brain tissue and thus others in the field can replicate the experiments.

The study also replicates the structure of the single protofilament using cryo-EM from Yang et al. and extends the findings to show that amplified DLB fibrils are a mixture of a single protofilament and two protofilament fibrils with a low twist.

This is a very important study for the field.

We appreciate this feedback from the reviewer.

Reviewer #2 (Remarks to the Author):

1. One of my MAJOR/MAIN Concerns with this work is highlighted in the data the authors now provide in Supplementary Note Figure 9. It shows that the samples show aSyn truncations that are seen to be present in the undigested 2nd Cycle samples and that increase significantly in the 4th Cycle undigested samples. In fact, for some samples, the amount of intact full-length aSyn represents only a minor species, e.g. 4th Cycle samples (LB1, LBD2, MSA2, and MSA3). Even in the other samples the ratio of ~50/50 of full-length to truncated samples. This raises several important questions about what is been amplified and whether mixtures of full-length and truncated amplified fibrils are formed. This has significant implications for the interpretation of the data, see below.

We described the Supplementary Notes Fig. 9 data in the results section (Results, page 6, lines 170-180) of the revised manuscript as follows:

“SDS-PAGE analysis of amplified fibrils indicated that proteolytic cleavage of A α yn occurred during the first four cycles of incubation, producing fragments with an approximately 2 kD reduction in size. This is likely produced by proteases present in insoluble fractions from brain tissue, and may correspond to previously observed proteolytic fragments of A α yn extracted from human brain tissue^{48,49}. The relative levels of proteolytic fragments were reduced substantially after 6 amplification cycles, which corresponded to a substantial decrease in non-resolved tissue-derived proteins relative to synuclein bands (Supplementary Notes Fig. 9).”

As we pointed out in our previous revision, this reduction in size from proteolytic cleavage likely corresponds to C-terminal proteolytic cleavage of A α yn present in fibrils extracted from Lewy body dementia (LBD) brain tissue. To confirm this, we performed western blots with antibodies that recognize a range of epitopes in A α yn protein. These new results indicate that the cleavage products lack C-terminal epitopes but not N-terminal epitopes (Supplementary Notes Fig. 10). The C-terminal cleavage events remove 15-30 amino acids from the C-terminus, consistent with the cleavage products present in LBD tissue-derived fibrils. Importantly, the cleavage sites occur in the disordered C-terminal region, and do not occur within regions of highly ordered beta sheet structure. Thus, the presence of C-terminal cleavage products reproduces a post-translational change present in A α yn fibrils derived from tissue, and is unlikely to produce heterogeneity in the beta sheet structure of fibrils. We updated the above paragraph (Results, page 5) as well as the figure legend for Supplementary Notes Fig 9 (Supplementary Notes, page 13) to further clarify this.

2. Panel C in Supp Notes Fig 9 show several inconsistencies and the WB does not suggest that these sample represent aggregated/amplified samples for the following reasons; 1) all the amplified samples from the 2nd and 4th Cycle without exception show the presence of high molecular weight species and band streaks, reflecting the presence of aggregated/fibrillar forms of aSyn; and 2) the presence of truncated species in significant amounts. None of these features is shown in the samples from the 6th amplification cycle, suggesting that the samples used here reflect primary soluble and not aggregated forms of aSyn. Consistent with this is the fact that all samples, including the controls, show a similar PK digestion band pattern, which is not expected, especially between LBD1 and MSA cases (Figure 1b and 1d). The SDS-PAGE profiles of the different cycles show that the fibrils obtained in each cycle have different biochemical compositions. Finally, the presence of truncated aSyn and these large variations from one cycle to another suggests that the procedure is not reproducible.

Supp Note Figure 9 MUST BE INCLUDED IN THE MAIN FIGURES of the revised version of the manuscript. Without this data, it is not possible for the reader to accurately interpret the results.

Supplementary Notes Fig. 9 is not a western blot (WB), but instead SDS-PAGE with SYPRO-Ruby staining to detect all proteins. For SDS-PAGE analysis, the insoluble proteins from the amplified fibril preparations were isolated by ultracentrifugation at 100,000 x g, and then solubilized with SDS-PAGE sample buffer. The higher molecular weight protein species in the 2nd and 4th cycle samples are predominantly proteins derived from the tissue insoluble fractions

used as seeds and not A β , as confirmed by our new western blot data shown in Supplementary Notes Fig. 10. The levels of these tissue-derived insoluble proteins relative to insoluble A β are substantially reduced with increasing amplification cycles as expected, which is why 6th cycle samples are predominantly composed of A β . We added an additional sentence to the figure legend for Supplementary Notes Fig. 9 (Supplementary Notes, page 13) to clarify this interpretation.

In our experiments with LBD amplified fibrils, most bands generated by proteinase K (PK) digestion are in the range of 10-15 kD. Therefore, we are not surprised by the lack of substantial differences between LBD and multiple system atrophy (MSA) fibril conformers. Since cleavages are in the range of 1-5 kD, they primarily occur in the disordered regions at the N- and C-termini and are therefore not directly related to differences in the highly ordered beta sheet region in the structural models derived from cryo-EM and SSNMR. We did observe one difference between LBD and MSA, a lower molecular weight (~7 kD) cleavage product present in PK digests of MSA fibrils but not LBD fibrils. We revised the text in the Results section (Results, page 6, lines 172-187) to clarify these points. The revised text is as follows:

“Western blot analysis demonstrated that these fragments were produced by cleavage at the C-terminus, likely by proteases present in insoluble fractions from brain tissue (Supplementary Notes Fig. 10). This C-terminal cleavage of A β that occurs during fibril amplification corresponds to a high percentage of C-terminal cleavages present in A β fibrils extracted from human brain tissue⁴⁸⁻⁵⁰. These C-terminal 30 amino acids are disordered and not part of the beta sheet structure in A β fibrils isolated from LBD and MSA postmortem tissue^{37,38}. The relative levels of truncated protein were reduced substantially after 6 amplification cycles. In addition, levels of tissue-derived proteins were also substantially reduced relative to A β (Supplementary Notes Fig. 9). Analysis of A β fibrils by proteinase K digestion showed distinct band patterns for IVTris fibrils compared to amplified fibrils, but relatively small differences in band patterns between different types of amplified fibrils. Proteinase K produced fragments with relative molecular weights predominantly in the range of 10 to 15 kD that were similar between LBD and MSA fibrils, which may be explained by the fact that these small cleavages of 1-5 kD occur in the disordered N- and C-terminal regions. However, a faint band at approximately 7 kD was visible in PK digests of MSA fibrils but not LBD fibrils.”

Importantly, in our previous revision, we added SSNMR data to demonstrate that fibril structure is consistent among multiple LBD samples from either the same case, or different autopsy cases.

We are open to including Supplementary Notes Fig. 9 and 10 as a main figure but this will exceed the journal's limit on the number of figures and tables. As it stands, we have worked to draw appropriate attention to this issue in the main text and refer readers to the Supplementary Notes Figures.

3. The band intensity for the amplified samples (Supp notes figure 6, Cycle 6) suggests high variability in the concentration of a β among all the samples, which could also influence the in vitro kinetics and cell seeding studies.

We provided quantification of the concentration of A α Syn fibrils amplified from 4 different cases in Supplementary Notes Table 1. These concentration measurements, which were determined by our radioligand binding assay, were validated with two additional orthogonal assays (A $_{280}$ and micro-BCA). We do observe different levels of fibril amplification from different cases. This is expected given the wide range of A α Syn fibril accumulation among different human postmortem autopsy cases and tissue samples (Ann Clin Transl Neurol. 9(2), 106-121, 2022). We utilized our concentration measurements to adjust the amounts of fibrils added in the cell seeding and fibril growth assays, and the consistency of this approach is evident in the data.

With respect to the variable band intensity in cycle 6 of the SDS-PAGE gels, this is likely related to variable losses during the preparation of very low amounts (300 ng) of A α Syn fibrils isolated by centrifugation for highly sensitive SYPRO-Ruby stain. Although we would prefer to show greater consistency of loading for this gel, the results are sufficient to demonstrate the relative levels of proteolytic cleavage products and tissue-derived proteins as discussed above.

4. Biosensor assays: The presence of distinct and variable amounts of full-length and truncated fibrils makes it difficult to interpret the seeding results using the biosensor assay. Are the seeding or morphological differences due to the different structure of the amplified full-length, truncated aSyn, or a combination of both? Difficult to answer. Furthermore, at the level of characterization provided, it is not possible to make any statement about similarities or differences in the morphological properties of the newly seeded aggregates. If they want to expand on this, please highlight these differences more clearly. They all appear filamentous. No quantification is shown in the main figure, plotted. This is important to allow a direct comparison of the seeding and biochemical data for each sample, WB.

We addressed the reviewer's questions about variable concentration and truncation in our response to comments 1-3 above. Since truncations affect only the disordered C-terminal region and equal concentrations of amplified fibrils were added to the biosensor cells, the differences in morphology and seeding efficiency between the LBD and MSA amplified fibrils groups indicate differences in structure. Quantification data is provided in a Supplementary Notes Table 7 due to limitations on the number of figures and tables in the main manuscript file.

5. The authors should assess the amplified sample from the MSA soluble fraction by WB to see if it contains truncated aSyn species or quantify the amount of soluble and aggregated aSyn species. This could provide some insight as to why they see seeding with the soluble MSA fraction.

We considered this type of analysis. However, assessment of truncated A α Syn species in the soluble fraction by western blot and quantification of the soluble fraction is unlikely to be helpful since the soluble fraction primarily contains unincorporated monomeric A α Syn, and only low levels of soluble oligomeric species.

6. Fibril growth experiments. Again, the interpretation of the kinetic data and growth rates is not complicated by the variable heterogeneity of the samples due to the presence of truncated forms of aSyn.

We addressed the reviewer's questions about truncated Asyn in our response to comment 1 above. Since beta sheet structure is consistent across multiple LBD samples in our SSNMR studies, and truncations are unlikely to directly affect beta sheet structure, they are not a major limitation in the interpretation of this data.

7. In the revised version, no PK digestion data were included as requested. Only data for the samples from Cycle 6 for which this reviewer questions whether they are indeed amplified samples and no differences are seen among the LBD, MSA, and control samples that reflect structural differences.

We did not attempt PK digestion for earlier cycles of amplification because it would not provide useful information beyond what is provided in the results for cycle 6. Since truncated species are abundant at early cycles, it will be difficult to interpret results of PK digestion. In addition, PK digestion provides very limited information about structure.

8. "We developed a new method to amplify Asyn fibrils from LBD postmortem tissue, enabling high-resolution analysis of fibril structure. 2D classification results from single-particle cryo-EM indicate a combination of single protofilament and two protofilament fibrils with very low twist. "

2D class averages and evidence for double filaments are only shown for LBD, no data were shown for MSA and control samples. This is important since their findings differ from those of a recent study showing primarily single protofilament fibrils. Could this be due to the presence of fibrils derived from full-length and truncated aSyn protein?

Single particle cryo-EM analysis of MSA and control amplified fibrils is beyond the scope of the studies in this manuscript. We addressed the reviewer's questions about truncated Asyn in our response to comment 1 above. Truncations are also present in fibrils derived from tissue and are unlikely to directly affect beta sheet structure.

1. All of the above does not support one of the authors' main claims that they "We developed a new method to amplify Asyn fibrils from LBD postmortem tissue, enabling high resolution analysis of fibril structure.

We provided new data and clarified the interpretation of our data to address the reviewer's concerns as outlined above, and are confident that the data and text support this claim.

Minor Comments

1. The authors should provide a more descriptive analysis of the differences in the morphological properties and diameters of the fibrils obtained in the different cycles. For example, 1) Supp Notes figure 3 compared to 1,2, 4, 5, 6, and 2) difference in diameter between amplification cycles 2 and 4 and amplification cycle 6. The data suggest the appearance of a second population of fibrils with diameters of ~13 nm.

For example, what is meant by distinct here "These fibrils appear distinct from all forms of amplified fibrils, based on diameter and twist"

We changed the above sentence to “For these fibrils, designated IV_{Tris}, the appearance of two twisting protofilaments was more clearly delineated by negative stain transmission electron microscopy (TEM) than it was for the various forms of amplified fibrils.” in results section (Results section, lines 126-128).

Very vague interpretation of the data. What information could be derived about each population, e.g., size to morphology correlation

“Diameter measurements for LBD amplified fibrils indicated a bimodal distribution after 6 cycles of amplification, with diameters ranging from 6 nm to 15 nm, suggesting two populations with overlapping diameter measurements. A bimodal distribution was not apparent for fibrils obtained

We provided the following interpretation of the diameter measurements in the revised manuscript (Results section, page 5, lines 153-164):

“Diameter measurements for LBD amplified fibrils indicated a bimodal distribution after 6 cycles of amplification, with diameters ranging from 6 nm to 15 nm, suggesting two populations with overlapping diameter measurements. A bimodal distribution was not apparent for fibrils obtained after 2 cycles or 4 cycles, which had a smaller mean diameter (Supplementary Notes Fig. 7). Amplified MSA fibrils and IV_{Tris} fibrils generally had a broad distribution of diameter measurement while control amplified fibrils had a narrower distribution (Supplementary Notes Fig. 8). Limited precision of diameter measurements and variable rotational orientation of fibrils on the EM grids may contribute to the variability of diameter measurements among different populations. Negative stain EM images showed that LBD amplified fibrils were primarily associated with amorphous tissue-derived material after 2 and 4 cycles. After 6 cycles, tissue components were rare and frequent fibril-fibril association was observed, in both parallel and crossed orientations.”

We are uncertain about the significance of “size to morphology correlations” and prefer not to over-interpret the diameter measurements, given the limited precision of negative stain TEM images and the fact that the apparent fibril diameter varies based on the orientation of fibrils on the grid surface. However, the diameter measurements do provide evidence of two populations of fibrils for LBD amplified fibrils as we describe above. We added text on page 6, lines 197-198 to link this observation from negative stain TEM to the combination of single and two protofilament fibrils observed on 2D class averages from single particle cryo-EM data.

Reviewer #3 (Remarks to the Author):

The authors have appropriately addressed the issues raised earlier and provided important additional data and information in the revised version.

We appreciate the comments from the reviewer.